# Neutrophil metalloproteinase driven spleen damage hampers infection control of trypanosomiasis

Hien Thi Thu Pham [1,2,3], Stefan Magez[1,3,4], Boyoon Choi[1,3,4], Bolortsetseg Baatar [1], Joohee Jung [5] & Magdalena Radwanska [1,2] ✉

Recent blood transcriptomic analysis of rhodesiense sleeping sickness patients has revealed that neutrophil signature genes and activation markers constitute the top indicators of trypanosomiasis-associated inflammation. Here, we show that *Trypanosoma brucei* infection results in expansion and differentiation of four splenic neutrophil subpopulations, including *Mki67*+*Birc5*+*Gfi1*+*Cebpe*+ proliferation-competent precursors, two intermediate immature subpopulations and *Cebpb*+Spi1+*Irf7*+*Mcl1*+*Csf3r*+ inflammation reprogrammed mature neutrophils. Transcriptomic scRNA-seq profiling identified the largest immature subpopulation by *Mmp8/9* positive tertiary granule markers. We confirmed the presence of both metalloproteinases in extracellular spleen homogenates and plasma. During infection, these enzymes digest extracellular matrix components in the absence of sufficient TIMP inhibitory activity, driving remodeling of the spleen follicular architecture. Neutrophil depletion prevents the occurrence of organ damage, resulting in increased plasma cell numbers and prolonged host survival. We conclude that trypanosomiasis-associated neutrophil activation is a major contributor to the destruction of the secondary lymphoid architecture, required for maintaining an efficient adaptive immune response.

*Trypanosoma brucei* is a unicellular flagellated parasite causing Human African Trypanosomiasis (HAT) or sleeping sickness[1]. This disease is characterized by a hemolymphatic stage, followed by a meningoencephalitic second stage with a deadly outcome[2]. Infection-associated pathologies are linked to inflammatory host responses, including lymphoadenopathy and hepatosplenomegaly[3]. During chronic infection, trypanosomes reside in the spleen, causing architectural disruption as a key driver in disabling B cell effector functions. The underlying mechanisms of spleen tissue pathology remain, however, largely unknown[4,5]. Likewise, questions remain with respect to the true drivers of HAT pathology in the context of brain inflammation, anemia, and tissue tropism. Therefore, the introduction of scRNA-seq analysis

and other high-resolution molecular technologies has sparked a renewed interest in trypanosomiasis[6–10].

Antigenic variation of variant surface glycoproteins (VSGs) helps trypanosome in immune evasion. The ability to switch antigenic coats, and the capacity to rapidly remove surface-bound antibodies, are both crucial for parasite survival[11–14]. To further increase infection success, trypanosomes ablate the host B cell compartment and express the invariant surface glycoprotein ISG65 that functionally neutralizes C3 complement[15–18]. While lymphocyte numbers decrease, myeloid cell numbers, and in particular neutrophil counts, increase throughout infection[19]. These cells are also recruited to the dermis after a tsetse bite[20].

[1]Laboratory for Biomedical Research, Department of Environmental Technology, Food Technology and Molecular Biotechnology KR01, Ghent University Global Campus, Incheon, South Korea. [2]Department of Biomedical Molecular Biology, Ghent University, Ghent, Belgium. [3]Laboratory of Cellular and Molecular Immunology, Vrije Universiteit Brussel, Brussels, Belgium. [4]Department of Biochemistry and Microbiology, Ghent University, Ghent, Belgium. [5]Duksung Women's University, Seoul, South Korea. ✉e-mail: magdalena.radwanska@Ugent.be

Neutrophils deliver the first line of the innate defense against invading pathogens by releasing reactive oxygen species (ROS), extracellular traps (NETs), and cytotoxic enzymes[21]. However, the latter can also contribute to the occurrence of tissue pathology. Prior to their release, these enzymes are stored in three types of granules[22]. Primary azurophilic granules contain myeloperoxidase (MPO) and serine proteases, including neutrophil elastase (NE) and cathepsin G. Secondary so-called 'specific' granules comprise the iron storage and transporter molecules lactoferrin and lipocalin 2, as well as anti-bacterial peptides. Tertiary gelatinase granules contain the metalloproteinases MMP-8 and MMP-9, which are able to digest collagen and elastin[23–26]. Interestingly, neutrophil *Mmp9* has been identified as the top differentially expressed gene (DEG) marker in early-stage HAT blood samples[27,28]. This is important as MMPs facilitate the recruitment of neutrophils to inflammation sites by cellular passage through the extracellular matrix (ECM)[29]. In secondary lymphoid tissues, this matrix normally provides a scaffold for immune cells, composed of collagen type I–IV, elastin, laminins, and heparan sulfate proteoglycans[30].

In recent years, the trypanosome literature has mainly focused on the role of neutrophils in parasite elimination. While in vitro assays with culture-form *T. brucei* parasites and neutrophils isolated from non-infected mice or cattle resulted in the release of NETs, in vivo studies rarely show the involvement of NETs in parasite elimination[20,31–33]. Ex vivo confocal imaging shows that after a tsetse bite, neutrophils are recruited to dermal sites, but their phagocytosis or NET formation is seldom observed. Moreover, neutropenia does not result in a significant increase of parasite numbers, showing a limited role for neutrophils in in vivo parasitemia control[20]. Given that neutrophil activation can result in collateral damage in host tissues, our focus here is now on the in vivo infection-associated expansion of neutrophils and its role in host pathology development.

Neutrophils are produced in vast numbers in the bone marrow, go through a series of precursors, and progressively mature and differentiate before being released into the bloodstream. Emerging mature neutrophils lose their proliferative capacity and exhibit downregulated transcriptomic profiles[34]. Recent scRNA-seq data analysis has allowed us to annotate different subpopulations, including pre-neutrophils (preNeus) as well as immature and mature neutrophils, of which the latter are no longer undergoing active cell division[35–37]. Our study shows the heterogeneity and alteration of neutrophil activation upon *T. brucei* infection. Combined results lead to the conclusion that neutrophil-driven damage to the secondary lymphoid organ architecture impairs the maintenance of a fully functional adaptive immune response against *T. brucei*.

## Results

### *T. brucei* infection triggers the proliferation, differentiation, and reprogramming of spleen neutrophils

During the early stage of trypanosome infection, granulocytes gradually expand in secondary lymphoid organs. As we have previously shown, spleen neutrophil numbers reach their peak two weeks after infection (14 dpi), coinciding with the occurrence of architectural organ destruction[15,19]. Now we demonstrate that the CD11b$^+$ Ly6C$^+$ neutrophil population shows a Ly6G/CD177 surface expression heterogeneity and a 16-fold increase in size (Fig. 1a, b, Supplementary Fig. s1). In bone marrow and blood, neutrophil heterogeneity is not affected to the same extent, and cell numbers only increase 1.3 to 2-fold. Hence, focusing on the spleen, cells from naïve and *T. brucei*-infected mice were subjected to scRNA-seq analysis. Merged data were dimensionality reduced using UMAP, followed by cell population annotation (Fig. 1c, Supplementary Fig. s2). Neutrophils were identified by canonical markers[5,6,35–39], including *Ly6g*, *Cd177*, *Ly6c2*, *Itgam* (CD11b), *Cxcr2* and *Cxcr4* (Fig. 1d). To enhance subpopulation exploration, neutrophils were extracted from the dataset and

subjected to a new dimensionality reduction and DEG analysis (Fig. 2a, Supplementary Fig. s3). For N1 and N2, the early granulopoiesis transcriptional factor *Cebpe* is a top-10 DEG marker. Both subclusters differ, however, in the expression of the cellular proliferation-related genes *Top2a* and *Pclaf*, which are switched off in N2. While both N3 and N4 neutrophils downregulate *Cebpe* expression, N3 upregulates transcription of tertiary granule proteinases *Mmp8*. N4 shows a shift towards transcription of chemokines/cytokines, including *Ccl6* and *Il1b*, characteristic of mature neutrophils. Using the Monocle computational trajectory analysis confirms a sequential pseudo-time N1–N4 neutrophil maturation process (Fig. 2b). This transcriptomic aging profile follows the same trend in both naïve and infected mice (Fig. 2c). While 14 dpi infected mice have increased spleen neutrophil numbers in all four subclusters, with N3 being the largest, the naïve sample contains predominantly mature (N4) neutrophils (Fig. 2d). Interestingly, this infection-induced spleen neutrophil profile remodeling is not mirrored in the bone marrow, where all 4 populations are present in both naïve and 14 dpi samples and undergo only minor changes during infection (Fig. 2e). Here, scRNA-seq data confirms the 1.3-fold cell number increase in response to infection, as observed previously by flow cytometry, (Fig. 1b, Supplementary Fig. s4a), with markers identifying all 4 neutrophil subpopulations being the same as those used for the spleen (Supplementary Fig. s4b, c)[40]. To gain in-depth insights into mechanisms of infection-driven spleen neutrophil granulopoiesis and differentiation, the expression of transcription factors from the C/EBP family was analyzed, showing that besides *Cebpe*, N1 neutrophils also express *Gfi1* and *Ms4a3* (Fig. 2f). These transcriptional factors are characteristic for proliferation-competent precursors pre-neutrophils (preNeu)[35,36]. In addition to the top DEG markers *Pclaf*, *Top2a* listed above, the N1 subcluster cells highly express *Ube2c* and *Ube2s* involved in DNA repair and replication. In the N2 subcluster, expression of *Cebpe* is maintained, but genes involved in cell cycle and proliferation are switched off, marking the differentiation into immature neutrophils. The N3 subcluster shows a decreased expression of both *Gfi1* and *Cebpe*, indicating continued differentiation of immature neutrophils. Finally, the profile of N4 neutrophils is distinguished by the high expression of *Cebpb*, *Cebpd*, and *Spi1* instead of *Cebpe*. These three transcriptional factors characterize late-stage granulopoiesis and infection-independent differentiation into mature neutrophils[35,36]. However, the infection does have a transcription reprogramming effect on other genes, with the DEG analysis between naïve and 14 dpi mature N4 neutrophils revealing changes in cellular responses to interferons (IFNs), such as the upregulation of *Stat1*. In addition, *T. brucei* induces upregulation of *Irf7 and Ifit3*, involved in type I interferon (IFN)-mediated inflammation, as well as *Ifi204*, *Ifi47*, *Isg15*, *Ifitm3*, *Igtp*, *and Gbp7*, known to mediate IFN-γ responses (Fig. 2f, Supplementary Table s4). In contrast, infection downregulates the expression of *Junb*, involved in transcriptional cell growth regulation, the *Jund* gene mediating differentiation, and the interleukin 1 receptor accessory protein *Il1rap* gene, implemented in a dampening of an excessive inflammation[41,42].

### Infection-induced neutrophils have an anti-apoptotic phenotype

Neutrophils are generally considered to have a very short half-life, undergoing rapid turnover resulting in cell death[43,44]. Hence, the high number of neutrophils residing in the spleen during *T. brucei* infection could indicate a delayed neutrophil apoptosis. This hypothesis is corroborated for the preNeu N1 subpopulation, as the Fig. 3a heatmap shows an infection-triggered expression of the apoptosis inhibitor *Birc5* (survivin) and the proliferation marker *Mki67*. Simultaneously, the mature N4 subpopulations express high levels of the cell survival regulator *Mcl1*, as well as *Csf3r*, coding for the G-CSF receptor. Figure 3b confirms that the overall anti-apoptotic gene expression score in the mature N4 cluster is significantly increased at 14 dpi. For

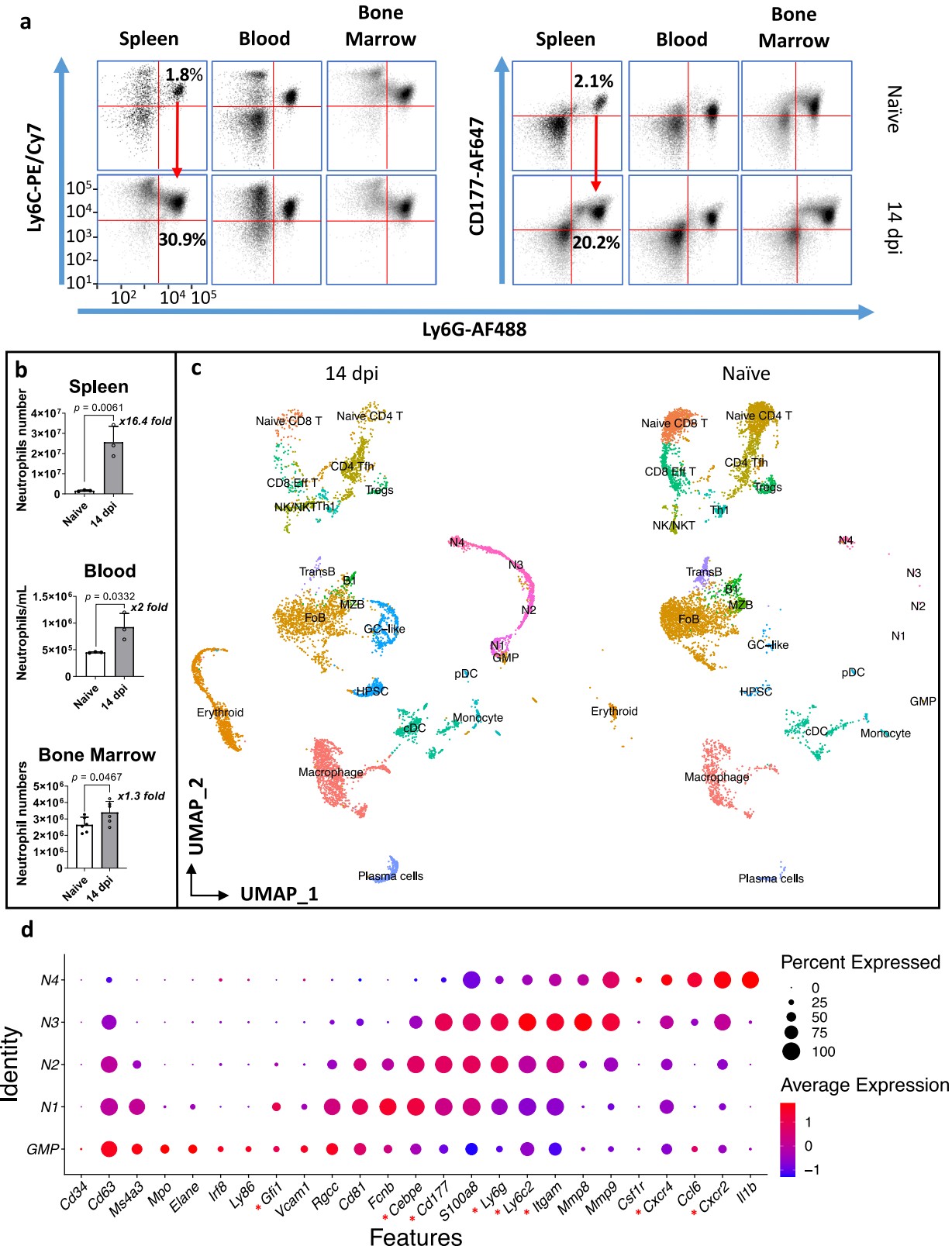

immature N2 and N3 neutrophils, elevated gene expression occurs for calprotectin (*S100a8/S100a9*), known to have a suppressive effect on neutrophil apoptosis under inflammatory conditions. Likewise, in the N4 subcluster *T. brucei* infection induces increased *S100a8/S100a9* expression. To corroborate this data, concentrations of calprotectin were measured by ELISA, showing that spleen extracellular homogenates of infected mice contain high levels of this heterodimer

(Fig. 3c). Infection does not affect the expression of pro-apoptotic marker genes (Fig. 3d, e), with all neutrophils being characterized by low expression of genes encoding various caspases (*Casp2, Casp3, Casp7, Casp8, Casp9, Casp12*) and pro-apoptotic regulators such as *Bax, Bad, Bbc3, Bcl2l11, Pidd1*, and *Bak1*. These data were confirmed by flow cytometry, where during infection, the proportion of caspase-3 positive spleen neutrophils did not increase (Fig. 3f).

**Fig. 1 | Identification of neutrophils during *T. brucei* infection. a** Altered surface expression of Ly6C, Ly6G, and CD177 on neutrophils of naïve mice (upper panel) compared to 14 dpi (lower panel). Spleen, blood, and bone marrow samples were analyzed by flow cytometry. Data illustrates profiles of one representative mouse out of three from one of four (SP, BM) or two (blood) independent experiments. The scaling of axes is identical for all plots. **b** Corresponding numerical changes in absolute neutrophil numbers in the spleen, blood, and bone marrow, isolated from naïve mice (white with black border) compared to 14 dpi mice (dark gray). Data represent means ± S.D. obtained with groups 3 mice (Spleen/Blood) or 6 mice (BM). Pairwise comparison was conducted using an unpaired two-tailed Student's *t*-test for analysis, with *p* < 0.05 considered as statistically significant. Source data are provided as a Source Data file. **c** Uniform Manifold Approximation and Projection (UMAP) plot displaying separate splenocyte data of 14 dpi and naïve mice, with cell types plotted by distinct colors. **d** Dot plot illustrating the average expression of neutrophil marker genes across each neutrophil subcluster (N1–N4) using combined data of 14 dpi and naïve mice, as well as the related granulocyte-macrophage progenitors (GMP), expressing a mix of monocytic fate-determining genes *Irf8 and Ly86* and combination of neutrophil progenitor genes *Elane, Mpo, Gfi1, Rgcc, Vcam1, and CD63*[40]. N1–N4 neutrophil subpopulations were annotated using neutrophil-specific *Gfi1* and *Cebpe* marker genes in combination with *Cd177, Ly6g, S100a8, Ly6c, Itgam, Cxcr4,* and *Cxcr2*[35–37,40]. Dot size indicates the percentage of cells within each subcluster expressing more than one read of the corresponding gene. Key marker genes are highlighted with an indicator (*).

Interestingly, *T. brucei* infection results in overall downregulation of CXCR2 neutrophil surface expression (Fig. 3g), counteracting the very significant increase of the migratory inducing receptor ligand CXCL2 concentration (Fig. 3h). In contrast, no modulation of CXCR4 was observed, despite the higher CXCL12 levels present in extracellular spleen homogenates of infected mice (Fig. 3h). Combined, this suggests that splenic neutrophil retention in the spleen relies on a CXCR2/CXCL2 interaction, rather than the CXCR4/CXCL12 axis.

### Infection triggers degranulation of enzymes into the tissue and bloodstream

Neutrophils respond to infection by producing antimicrobial ROS involving the NADPH oxidase complex. While gp91 (*Cybb*) and gp21 (*Cyba*) constitute the catalytic core of the complex, p40 (*Ncf4*), p47 (*Ncf1*), p67 (*Ncf2*), and the Rac2 protein (encoded by *Rac2*) are regulatory components required for enzyme activity. Interestingly, *T. brucei* infection triggers a significant increase of the overall NADPH oxidase complex expression score in the mature N4 neutrophils (Fig. 4a), however none of the 6 genes listed above shows an individual significant alteration of expression as a result of infection by itself (Fig. 4b). In addition, expression of the *Cyba* and *Cybb* genes is elevated during infection in preNeu and both immature subpopulations, as compared to mature neutrophils. Other genes encoding the regulatory complex exhibit varying expression levels, with 14 dpi N4 neutrophils showing an occasional individual high expression of *Ncf2* as well as *Rac2* in conjunction with *Rac1*, indicative of granule release (Fig. 4b).

Functional analysis of infection-induced neutrophils was further performed based on transcriptome analysis of various enzymes (Fig. 4c). Here, 14 dpi preNeu N1 cells show occasional high expression of *Mpo* and *Elane*. Both genes encode azurophilic granules enzymes, with MPO being an NADPH oxidase-linked enzyme important for pathogen elimination and *Elane* coding for the elastase NE required for the formation of NETs. The gene expression pattern of the N2 immature subpopulation shows commitment to transcription of *Lcn2* and *Ltf* genes, both encoding proteins involved in iron transport. The large N3 subpopulation of immature neutrophils upregulates gene expression of *Mmp8* and *Mmp9*, encoding matrix metalloproteinases associated with gelatinase granules. Expression of chemokines and cytokines genes is overall low, except for *Il1b*, a known marker for mature neutrophils. The latter is, however, independent of infection, as is the expression of genes involved in phagocytosis (Supplementary Fig. s5).

To corroborate the relevance of the observed granule-specific gene expression profiles, the infection-induced presence of myeloperoxidase (MPO) was assessed by immunohistochemistry (Fig. 4d). Results confirm the intracellular presence of this enzyme during *T. brucei*-infection, as MPO⁺Ly6G⁺ neutrophils formed clear foci. In parallel, free MPO was measured in extracellular spleen homogenates, showing increased levels at 14 dpi (Fig. 4e). In contrast, plasma MPO levels for both experimental groups remained below the detection limit, indicating that MPO is released locally in the spleen. Next, spleen sections were stained using anti-Ly6G and anti-elastase antibodies.

Overall, only very few neutrophils were positive for elastase in the tissue (Fig. 4d). However, high levels of the free enzyme were detected in extracellular spleen homogenates and plasma of infected mice, pointing towards a systemic origin (Fig. 4e). Spleen sections were also screened for the presence of NETs by staining for histones in decondensed chromatin. This analysis shows no formation of fine or thick extracellular chromatin strands upon infection, corroborating the absence of adequate levels of available intracellular elastase required for NET formation. Extracellular spleen homogenates and plasma were also checked for the presence of lipocalin-2 and lactoferrin, showing increased levels of both in the spleen homogenates of infected mice. In contrast, only an infection-induced increase in lipocalin-2, but not lactoferrin, was observed in the plasma of infected mice (Fig. 4e).

### Neutrophil expansion results in severe MMP-driven ECM remodeling and destruction of B cell follicles

*T. brucei* infection triggers the release of the zinc-dependent matrix metalloproteinases MMP-8 and MMP-9 from gelatinase granules, measurable in large quantities in extracellular spleen homogenate and plasma (Fig. 5a). The presence of higher levels of both enzymes in the plasma compared to homogenates suggest an additional systemic production of these molecules, as a result of the overall inflammation that accompanies *T. brucei* infection. Much lower concentrations of MMP-25 were recorded. ScRNA-seq analysis shows that in the spleen, these three metalloproteinases are uniquely expressed by neutrophils (Fig. 5b) and are listed in the top 10 DEGs for neutrophil subpopulation clustering (Supplementary Fig. s3a), while other MMPs were absent. Interestingly, detailed expression analysis of these *Mmps* also shows that the infection-induced elevated enzyme levels observed in ELISA (Fig. 5a) are mostly the result of an increased number of *Mmp*-expressing cells rather than an augmented expression at the individual cell level. However, increased MMP-8 presence during infection is the result of both, with the major N3 subpopulation showing the highest level of expression and the N4 mature neutrophils showing increased infection-induced expression as well as a cell number expansion (Fig. 5c, Supplementary Table 4).

While the release of MMPs facilitates the migration of neutrophils to the inflammation sites, their excessive production can cause severe tissue pathology through the remodeling of the collagen and elastin-containing extracellular matrix (ECM)[45,46]. As MMP-8 and MMP-9 digest collagen and MMP-9 also degrade elastin, *T. brucei*-induced ECM remodeling can be demonstrated using specific antibody tissue staining combined with hematoxylin/eosin staining (Fig. 5d). Combined data show the profound destruction of the spleen architecture during infection, resulting in the loss of follicles.

Under regular immunostimulatory conditions, spleen follicles allow the generation of effector plasma cells (PCs) that can secrete high-affinity antibodies. Hence, given the severity of trypanosomosis-associated spleen architectural destruction, a NicheNet analysis was performed to assess the effect on cell–cell interactions between infection-induced neutrophils and PCs. This reveals a major crosstalk with immature N2 neutrophils as the primary effector cells (Fig. 6a)[47].

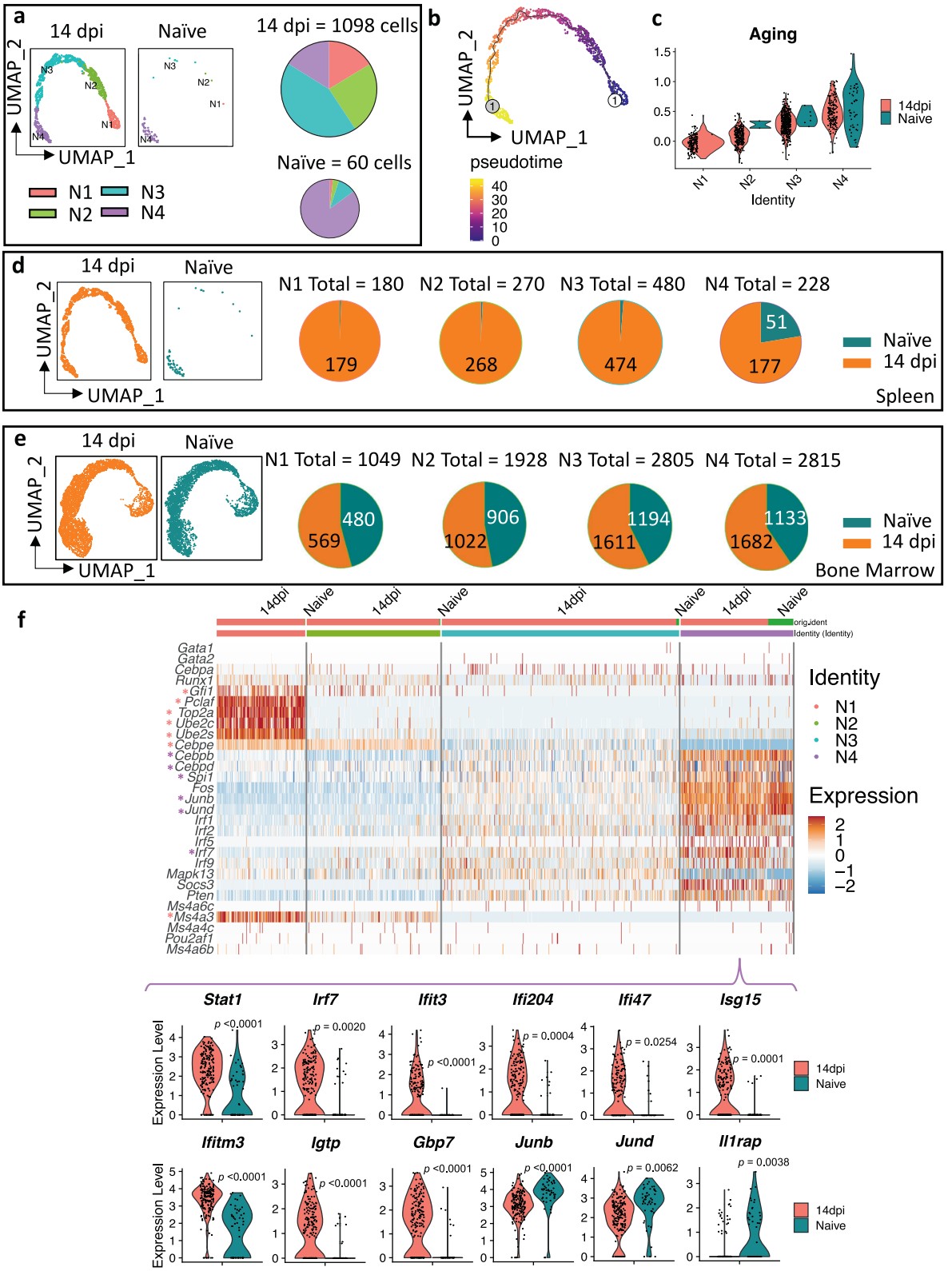

Next, genes encoding receptors on PCs identified by NicheNet were subjected to a DEG analysis, comparing naïve and 14 dpi samples (Fig. 6b), showing downregulated expression of (i) *Tnfrsf17* (BCMA), a B cell maturation antigen responsible for delivering a pro-survival signal to terminally differentiating B cells[48] and (ii) *Tnfrsf13b* (TACI), a transmembrane activator and CAML interactor, known to promote B cells differentiation and survival of PCs[49]. Simultaneously, PCs

upregulate the expression of Spn, encoding the CD43 receptor involved in cell adhesion and interactions with E-selectins and galectins (*Lgals3*), linked to B cell−PC differentiation[50]. Prior interaction potential analysis between PC receptors and neutrophil ligands confirms the involvement of the BAFF (*Tnfsf13b*) signaling pathway (Fig. 6c). While infection-induced spleen neutrophils produce and express high levels of BAFF, PCs downregulate the corresponding

**Fig. 2 | Infection triggers the proliferation, differentiation, and reprogramming of spleen neutrophils. a** Re-clustered UMAP plot after a new dimensionality reduction, using 1158 neutrophils extracted from Fig. 1c, yielding four discrete subclusters: N1 (pink), N2 (Green), N3 (blue), and N4 (purple). Pie charts show the proportions of the four subclusters for 14 dpi and naïve control samples, scaled to the total number of barcoded cells indicated for each sample. **b** Monocle trajectory analysis using UMAP plot embedding of combined neutrophils of 14 dpi mice and naïve samples, colored for pseudo-time (dark blue color denotes the starting point, yellow depicts the final differentiation stage, and colors in between show different transition stages of cells). **c** Violin plots representing the average scaled expression scores of marker genes related to aging[37] in N1–N4 subclusters of 14 dpi mice (red) and naïve (dark blue) mice. **d** Re-clustered UMAP plot after a new dimensionality

reduction of neutrophil populations in 14 dpi (orange) and naïve mice (dark blue), extracted from Fig. 1c. Pie charts demonstrate numbers of neutrophils in 14 dpi and naïve samples, making up for each of the four subclusters (N1–N4). **e** Similar to above, but for bone marrow, showing the overall lack of a substantial infection-driven change of BM neutrophil populations. **f** Row-scaled expression of genes encoding signature transcription factors[35], involved in granulopoiesis, differentiation, and proliferation per each cluster (N1–N4) in 14 dpi and naïve mice. Accompanying violin plots representing differences in gene expression of N4 mature neutrophils between 14 dpi (red) and naïve (dark blue) mice. For comparisons of individual genes, DEG analysis from the Seurat package was used, and p-values were extracted from the DEG list, with $p < 0.05$ considered statistically significant. Key marker genes are highlighted with a color-coded indicator (*).

receptors, in particular BCMA (Fig. 6d). Interestingly, the analysis of interactions between expanding immature neutrophils and Follicular B cells (FoBs) also resulted in identification of BAFF signaling as the main communication pathway (Fig. s6a), with infection-derived FoBs downregulating *Tnfrsf13b* (TACI), despite the upregulation of *Tnfsf13b* (BAFF) and *Tnfsf13* (APRIL) by N1 and N2 neutrophils (Supplementary Fig. s6b). Other downregulated DEGs in FoBs include *Tgfbr2* (Transforming Growth Factor beta receptor 2) and *Ptprc* (Protein Tyrosine Phosphate receptor type C− CD45). However, these interactions were not exclusive to B cells and neutrophils. Upregulated genes in FoBs encoded intracellular proteins related to ribosomal function, such as *Rack1* (intracellular protein kinase C receptor) and *Rpsa* (40S ribosomal protein) (Fig. s6b). Prior interaction potential analysis between FoBs and neutrophils also indicates strong crosstalk between *Icam2* and the corresponding *Itgam* and *Itgb2* genes, encoding adhesion molecules (Supplementary Fig, s6c)[51,52]. Overall, this combined cell–cell crosstalk analysis points to an infection-induced downregulation of the BAFF signaling pathway between B cells and expanding immature neutrophils. This corroborates the inhibition of pro-survival signals on the B cell targets as a major pathological event during progressing *T. brucei* infections[15].

### Neutrophil depletion prevents ECM damage via TIMP-dependent inhibition of MMPs

To better understand the mechanisms involved in neutrophil-driven organ pathology, a neutrophil depletion study was performed. Anti-Gr1 antibody treatment was started two days before *T. brucei* infection and followed by two-day interval treatment till 12 dpi. This results in significant depletion of neutrophils from the spleen and the bone marrow, with only a minor effect on other Ly6C+ monocytes (Fig. 7a, b). During infection, neutrophil depletion prevents *T. brucei*-induced collagen and elastin breakdown and thus avoids severe ECM remodeling (Fig. 7c). This coincides with the significant reduction of MMP-8 and MMP-9 levels in both extracellular spleen homogenates and plasma of infected mice (Fig. 7d).

Under homeostatic conditions, excessive MMP activity is counteracted by upregulated production of tissue inhibitors of metalloproteinases (TIMPs). Transcriptome analysis of naïve and 14 dpi neutrophils shows differential expression of *Timp2*, but not *Timp1*, during *T. brucei* infection (Supplementary Fig. s7a). At the level of the proteins, TIMP-1 and TIMP-2 concentrations are much higher in plasma than in spleen homogenates, indicating an additional systemic origin of these inhibitors rather than being produced by spleen neutrophils only. Neutrophil depletion significantly reduces *T. brucei*-induced TIMP-1 and TIMP-2 plasma concentrations as well as the TIMP-1 concentration in spleen homogenates. (Supplementary Fig. s7b). As TIMPs are biological MMP inhibitors, the ratio between these molecules is an indicator of progressing pathology and a predictor of inflammation-triggered ECM damage[53,54]. Here, data confirm that lower MMP-8/TIMP-1 and MMP-8/TIMP-2, as well as MMP-9/TIMP-1 and MMP-9/TIMP-2 values in extracellular spleen homogenates, coincide with reduced tissue damage as a result of the beneficial effect of neutrophil

depletion during *T. brucei* infection (Fig. 7e, f). The same is observed for MMP-8/TIMP-1, MMP-8/TIMP-2, and MMP-9/TIMP-1 in plasma. Combined, this indicates that the lack of sufficient systemic TIMP production needed to counterbalance neutrophil MMP-8 and MMP-9 secretion is a key determinant in the occurrence of *T. brucei*-induced spleen architecture destruction.

### Neutrophil depletion triggers PC generation and prolongs the life span of infected mice

Neutrophil depletion does not affect the T cell and NK1.1+ bone marrow or spleen compartment during infection (Fig. 8a and Supplementary Fig. s8). Likewise, neutrophil depletion also did not affect the previously described infection-induced reductions in bone marrow pre-Pro, pre- and pro- and Immature B cell numbers, or Marginal Zone B (MZB) and Follicular (FoB) B cell numbers in the spleen (Fig. 8b and Supplementary Fig. s9a). However, neutrophil depletion has a significant positive effect on the induction of spleen PCs during infection (Fig. 8b and Supplementary Fig. s9b). Staining of thin spleen tissue sections with anti-IgG FITC confirmed these observations, showing an increased number of IgG-positive B cells in infected mice after neutrophil depletion (Fig. 8c). This improved IgG+ PC formation during infection upon reduction of neutrophil numbers results in improved parasitemia control and significantly prolonged host survival during *T. brucei* infection (Fig. 8d, e).

## Discussion

*T. brucei* parasites cause infections characterized by hepatosplenomegaly[2,3] with excessive expansion of the spleen neutrophil compartment[19]. Here, we show that this process coincides with a distinct cell differentiation pattern normally confined to the bone marrow, with the expanding spleen neutrophil population containing proliferation-competent preNeu precursors that differentiate into immature and mature neutrophils. During infection, these cells can mediate severe ECM remodeling and breakdown. Interestingly, neutrophil depletion results in improved *T. brucei* control and prolonged host survival. This might appear counterintuitive, given the crucial role of neutrophils in the innate immune defense response[21], and prompted our investigation into the role of these cells in immunopathology. Indeed, diverting neutrophil activity could constitute a parasite evasion strategy. Trypanosomes have already been shown to exhibit multiple adaptive immunity evasion mechanisms[11–16]. However, so far, the modulation of innate cells, favoring infection progression, has not received major attention. Based on transcriptomic profiling, we demonstrate that *T. brucei* infection first triggers the proliferation of the spleen preNeu precursor population (N1), characterized by *Gfi1* and *Cebpe* expression, encoding early-stage granulopoiesis transcriptional factors. This suggests the occurrence of spleen emergency granulopoiesis in response to infection. N1 cells also express *Mpo* and *Elane*, marker genes for azurophilic granules, and *Lcn2* and *Ltf*, genes coding for iron transport proteins present in specific granules. The expansion of spleen preNeu cells has previously been demonstrated in models of sepsis and pancreatic carcinoma[35–37]. While these studies

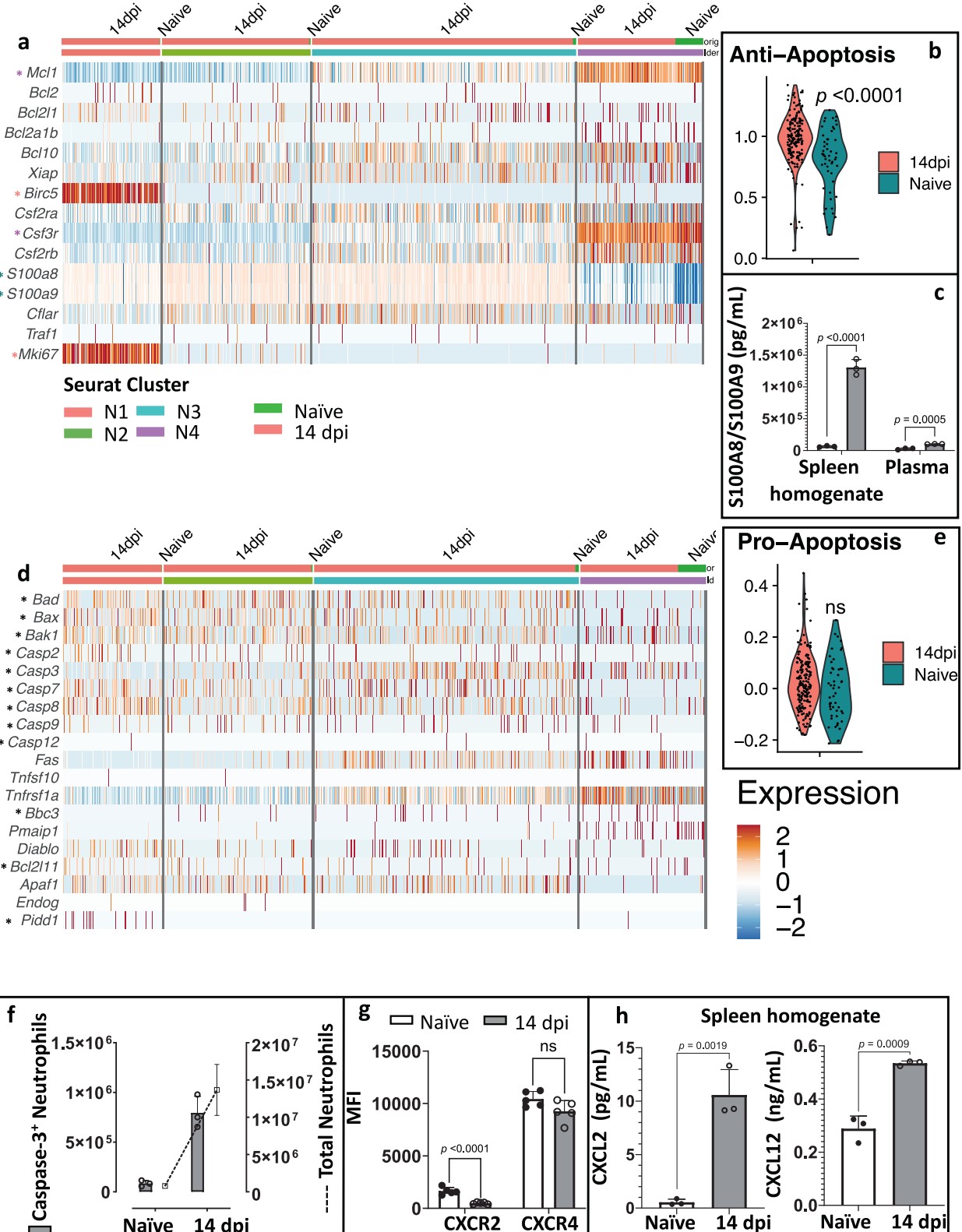

**Expression**

Seurat Cluster
N1  N3  Naïve
N2  N4  14 dpi

mainly focused on unraveling transcriptomic profiles, our current data goes beyond the in silico data analysis, detecting the intracellular and extracellular presence of these enzymes and associated functions. Results show that infected mice have increased numbers of spleen MPO⁺Ly6G⁺ neutrophils but very few NE⁺Ly6G⁺ neutrophils. Simultaneously, high concentrations of free MPO and NE were measured in spleen extracellular homogenates, indicating the degranulation of azurophilic granules. This shows that *T. brucei* infection triggers the systemic release of NE, coinciding with the breakdown of the elastin present within the ECM. Both MPO and NE were previously shown to participate in NET formation. However, during *T. brucei* infection, no sign of the release of either thin or thick DNA fibers was observed. This result might be attributed to the lack of increased intracellular NE concentration[55-57]. This observation also corroborates previous in vivo

**Fig. 3 | Infection-induced neutrophils show an anti-apoptotic phenotype.**
**a** Row-scaled gene expression of anti-apoptotic marker genes[35–37] in four neutrophil subclusters (N1–N4) of 14 dpi and naïve mice. **b** Average scaled expression score of anti-apoptotic marker genes in N4 mature neutrophils of 14 dpi (red) and naïve (dark blue) mice. **c** Concentrations of the S100A8/S100A9 complex in spleen extracellular homogenates and plasma. Results represent means ± S.D. of 3 mice per experimental group in one of three representative experiments, using an unpaired two-tailed Student's *t*-test for analysis, with *p* < 0.05 considered as statistically significant. **d** Row-scaled gene expression of selected pro-apoptotic marker genes[35–37] in N1–N4 neutrophils of 14 dpi and naïve mice. **e** Violin plots representing the average scaled expression score of pro-apoptotic marker genes in N4 mature neutrophils of 14 dpi (red) and naïve (dark blue) mice. **f** Number

of Caspase-3⁺ neutrophils (Y1-axis) and total neutrophil numbers (Y2-axis) in naïve and 14 dpi samples, showing one representative experiment with 3 mice per group (means ± S.D.). **g** Mean fluorescence intensity (MFI) of CXCR2 and CXCR4 on Ly6C⁺ Ly6G⁺ neutrophils (means ± S.D.) of combined results of 5 mice from two of three separate experiments. **h** CXCL2 and CXCL12 concentrations (means ± S.D.) in spleen extracellular homogenates of one of two (CXCL2) or three (CXCL12) validation experiments, with 3 mice per experimental group. For **b, e, g, h**, unpaired two-tailed Student's *t*-tests were used for comparison between experimental groups, with ns indicating non-significant differences for *p* ≥ 0.05 and *p* < 0.05 considered as statistically significant. Samples of naïve mice are indicated in white with a black border and 14 dpi mice in dark gray. Key marker genes are highlighted with a color-coded indicator (*). Source data are provided as a Source Data file.

data, showing that highly motile flagellated trypanosomes are rarely engulfed by phagocytic cells or trapped by NETs[20]. As trypanosomes escape these elimination processes, neutrophils appear to have a limited function in parasitemia control. This conclusion is now validated by our neutrophil depletion studies, and moreover evokes the question: why does neutrophil reduction in the spleen improve the overall parasitemia control? The answer to this could be linked to the content of the gelatinase granules and the infection-induced expansion of neutrophils, hardly present in the spleens of non-infected animals. Indeed, during infection, high levels of MMP-8 and MMP-9 are present in extracellular spleen homogenates and plasma. The genes encoding these proteases are significantly upregulated in expanding immature neutrophils. Interestingly, the detailed analysis of differential gene expression indicates that this subpopulation consists of the two subclusters N2 and N3. The N2 subset of immature neutrophils retains *Cebpe*, *Lcn2*, and *Ltf* expression but downregulates *Gfi1*, *Ms4a3*, *Mpo*, and *Elane* marker genes, as compared to preNeu precursors. In contrast, the N3 gene expression pattern is characterized by the downregulation of transcriptional factors, *Mpo* and *Elane*, and a shift towards high expression of *Mmps*. Both N2 and N3 transcriptomic profiles differ from N4 mature neutrophil subpopulations, with the latter showing the characteristic upregulation of genes *Cebpb*, *Cebpd*, and *Spi1*, associated with the regulation of late-stage granulopoiesis, in addition to the *Il1b* marker gene. Neutrophil subpopulations follow a differentiation pattern that, under normal conditions, takes place in the bone marrow (BM) and can be projected onto a single continuum, progressing from preNeu to immature neutrophils and subsequently to the final mature stage that is released to the periphery[35–37]. During trypanosome infection, this BM process continues to operate, and only minor changes in neutrophil composition are observed. In healthy spleens of non-infected mice, the majority of neutrophils are mature cells, while the two immature subpopulations and preNeus are virtually absent. However, our results show that trypanosome infection triggers the distinct emergence of a spleen neutrophil differentiation pattern that follows the same single continuum normally present in the BM. In addition, while mature neutrophils usually undergo senescence and are removed from inflammation sites, we show that during *T. brucei* infection, these cells exhibit delayed apoptosis. This involves high expression of *Mcl1*, a known regulator of cell survival, and the *Csf3r* gene, coding for the growth receptor for granulocyte colony-stimulating factor G-CSF. In addition, low expression of *Bax* is associated with prolonged survival and accumulation of neutrophils[58,59]. We also show that *T. brucei* infection reprograms mature neutrophils, giving this subpopulation a distinct gene expression pattern as compared to mature neutrophils from non-infected mice. Indeed, in contrast to the latter, infection-derived cells show upregulation of inflammation-specific transcriptional factors such as the *Irf7* gene, encoding a key transcriptional regulator of type I interferon (IFN)-dependent immune response and genes known to mediate IFN-γ responses. Neutrophils were shown to be responsive to type I interferon, and an *Irf7*-associated hyperactivation, similar to the profile observed here, has been recently described in patients suffering from COVID-19[60]. Interestingly, this recent finding

has triggered a therapeutic strategy, targeting neutrophils to reduce the severity of viral infections.

Our results show that during *T. brucei* infection, spleen neutrophil proliferation and differentiation are linked to the destruction of the ECM and overall splenic architecture. This includes severe remodeling of the white pulp containing lymphoid tissue structures such as follicles. We propose that the underlying destructive mechanism is the excessive extracellular release of MMP-8/MMP-9 as well as elastase, capable of breaking down the protein components of the ECM. This happens in the absence of sufficient TIMP inhibitory action, known to counteract developing organ pathology[61]. These data complement previous findings, showing that hemolymphatic stage HAT patients exhibit high expression of neutrophil marker genes *Mmp9* and *Cd177* in peripheral blood mononuclear cells (PBMCs). High concentrations of MMP-9 were also found in cerebrospinal fluid when trypanosomes cross the blood–brain barrier and invade the central nervous system[27,28]. While MMP-9 is able to break down type I and II–IV collagens, elastin, aggrecans, and laminins, MMP-8 cleaves collagen fibrils[62]. Under homeostatic conditions, excessive production of both MMPs is inhibited by TIMPs[61]. However, our data show insufficient compensatory TIMP-1 and TIMP-2 production during *T. brucei* infection. Interestingly, at the same time, high levels of extracellular NE are known to activate MMP-9 and cleave ECM proteins such as fibronectin, elastin, proteoglycans, and collagens[63,64]. As such, we conclude that trypanosomiasis-associated neutrophil activation is a major contributor to the destruction of the ECM, leading to damage of the secondary lymphoid follicular architecture required for maintaining an efficient adaptive immune response. This conclusion is supported by our finding that neutrophil depletion results in a reduction of ECM damage and B cell follicle destruction, two pathological events that we have described earlier as hallmarks of early trypanosome infection[15]. Consequently, compared to *T. brucei*-infected control mice, neutrophil-depleted infected mice have significantly higher numbers of differentiated CD138⁺ PCs in the spleen. Interestingly, CD138 mediates PC cell adherence to insoluble and soluble components of the 3D collagen structure of the ECM[65–67]. Hence, prevention of ECM damage may explain the observed PC spleen retention.

Severe infection-induced remodeling of the ECM impacts cell migration and homing to pre-defined compartments, as under homeostatic conditions, this matrix provides a non-cellular scaffolding to which cells attach. As such, the ECM is crucial for maintaining the integrity of secondary lymphoid organs and supporting organ compartmentalization. The role of the ECM is particularly important in the spleen, where T cells and B cells are disposed of in a highly organized manner, with B cells concentrated in the follicles. Therefore, the ECM is known to facilitate cellular migration and the interactions between activated B cells, follicular DCs (FDCs), and CD4+ T follicular helper cells (Tfh)[68]. This is crucial for the formation of germinal centers, where the process of somatic hypermutation (SHM) resulting in B cell affinity maturation, as well as class switch recombination (CSR), takes place, all required for high-affinity antibody production by PCs[69]. Analysis of cell-cell interactions between neutrophils and PCs showed that during

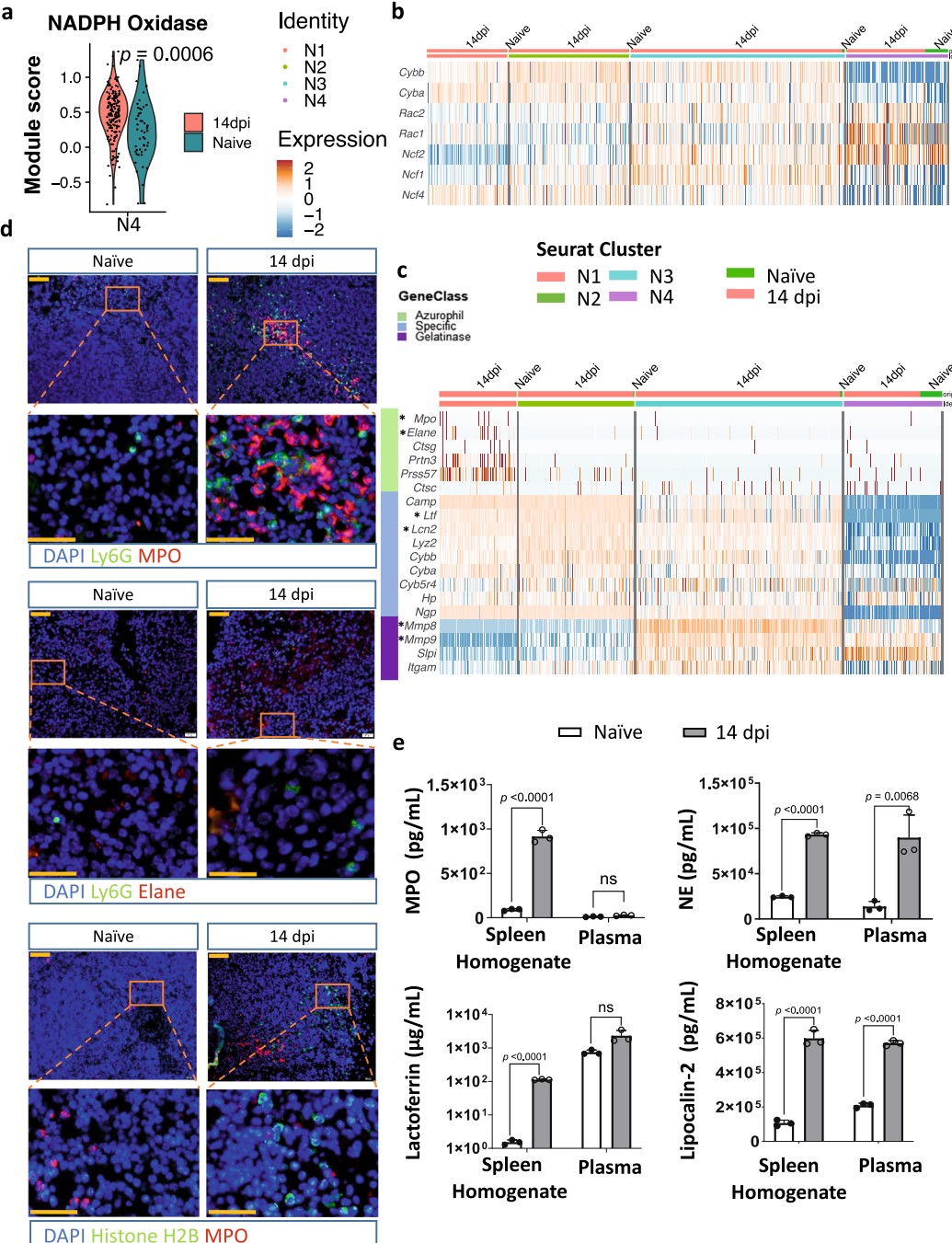

**Fig. 4 | Infection triggers degranulation of neutrophils. a** Violin plot representing the average scaled expression scores of genes related to NADPH oxidase in N4 mature neutrophils of 14 dpi (red) and naïve (dark blue) mice. **b** Heatmap showing row-scaled gene expression of selected marker genes[35–37] involved in activation of NADPH oxidase per cluster in 14 dpi and naïve mice. **c** Row-scaled expression of selected genes[35–37] encoding molecules in granules (azurophilic, specific, gelatinase) per cluster in naïve mice and 14 dpi mice. Key marker genes are highlighted with an indicator (*). **d** Representative immunofluorescence images of frozen thin spleen sections stained with anti-Ly6G (green) and anti-MPO (red) or Ly6G (green) and anti-NE (red). Tissue sections were also stained with anti-Histone H2B (green) and MPO (red). For all sections, the cell nucleus was visualized by DAPI staining (blue). The lower panel shows magnified images of the upper ones. Scale

bars equal 50 μm (upper) and 20 μm (lower). For each condition, spleen sections from three individual mice per experimental group were stained, showing one representative result. **e** Concentrations of MPO, NE, lactoferrin, and lipocalin-2 were measured in spleen extracellular homogenates and plasma by ELISA. Data represent means ± S.D. from one of three representative experiments with 3 mice per experimental group. Samples of naïve mice are indicated in white with a black border and 14 dpi mice in dark gray. A pairwise comparison was conducted between 14 dpi and naïve using an unpaired two-tailed Student's $t$-test for analysis, with ns indicating non-significant differences ($p \geq 0.05$) (for lactoferrin) or the estimated concentrations were below the detection range of the assay (for MPO), and $p < 0.05$ considered as statistically significant. Source data are provided as a Source Data file.

infection, the latter downregulate key survival receptors such as BCMA (*Tnfrsf17*) and TACI (*Tnfrsf13b*). These receptors belong to the family of TNF receptors that play a crucial role in B cell development[48,49]. Hence, infection-induced destruction of splenic ECM integrity may undermine

the hosts' ability to systemically eliminate the parasite due to the downregulation of survival signals on PCs in secondary immune organs. Interestingly, the homeostatic crosstalk between microglia and PCs, also involving BAFF signaling, was previously shown to counteract

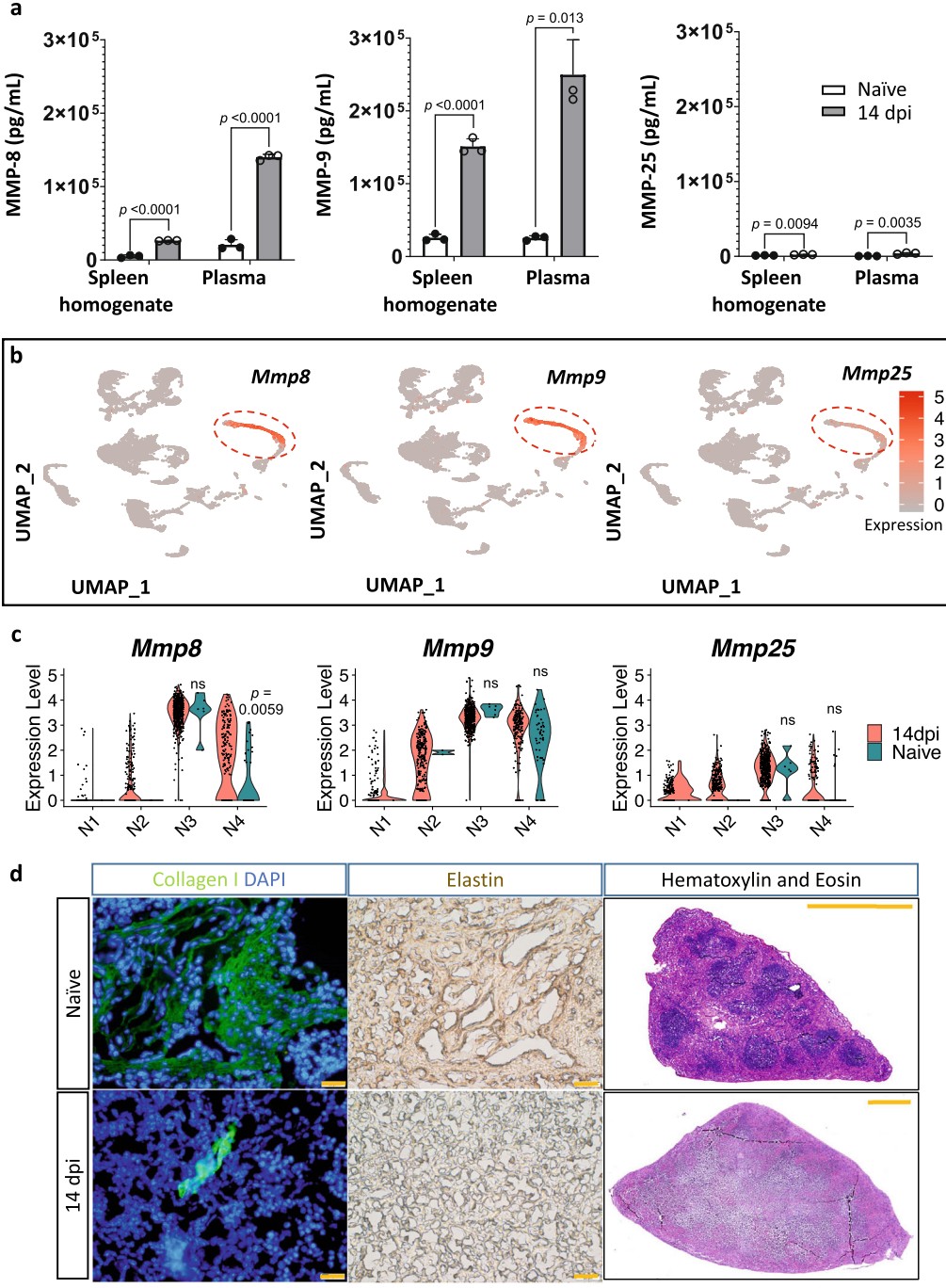

**Fig. 5 | Destruction of spleen architecture and ECM remodeling during *T. brucei* infection. a** Concentrations of MMP-8, MMP-9, and MMP-25 in spleen extracellular homogenates and plasma of naïve mice (white with black border) and 14 dpi mice (dark gray) were measured by ELISA. Data shows means ± S.D. from one of three (MMP8 and 9) or two (MMP25) validation experiments, with 3 mice per experimental group. An unpaired two-tailed Student's *t*-test was used for comparison, with ns indicating not significant for $p \geq 0.05$, and $p < 0.05$ considered as statistically significant. Source data are provided as a Source Data file. **b** UMAP feature plot demonstrating the expression pattern of *Mmp8*, *Mmp9*, and *Mmp25* genes, using combined splenocyte data from 14 dpi and naïve mice. **c** Violin plots representing differences in gene expression of *Mmp8*, *Mmp9*, and *Mmp25* between 14 dpi (red) and naïve (dark blue) mice. For comparisons of individual genes, *p*-values were used from the differentially expressed gene (DEG) analysis in Seurat. The Wilcoxon rank sum test was used for statistical analysis with *p*-values adjusted using the Bonferroni correction, with ns indicating not significant for $p \geq 0.05$, and $p < 0.05$ considered as statistically significant. **d** Representative immunofluorescence images of frozen thin spleen sections stained with anti-collagen type I (green) and DAPI (blue) for nucleus staining. Immunohistochemistry was also used to stain elastin fibers with anti-elastin antibodies and a secondary HRP-coupled detecting antibody (brown). Thin spleen sections were also stained with Hematoxylin and Eosin (H&E). Scale bars, 20 μm (collagen and elastin) and 1 mm (H&E). For each condition, spleen sections from three individual mice (each with two sections) per experimental group were stained, showing one representative result.

progressing trypanosomiasis-associated neuroinflammation[6]. Therefore, the BAFF signaling pathway seems to have an important regulatory function on cell survival in different organs during trypanosome infection.

In conclusion, we provide here the transcriptomic profiling of bone marrow and spleen neutrophil subpopulations and show that the latter expand and differentiate during *T. brucei* infection. These include proliferation-competent precursor preNeus, as well as

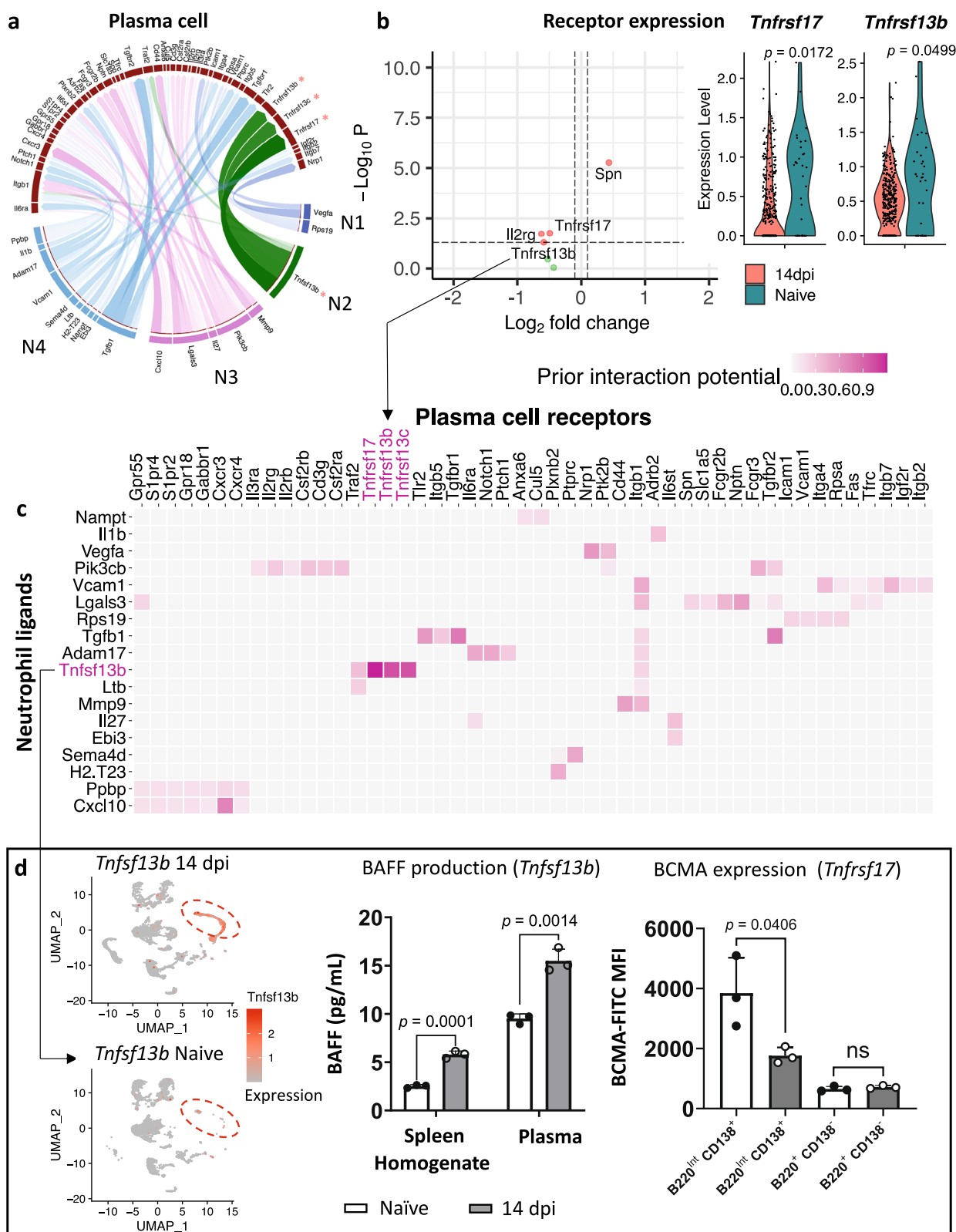

immature and inflammation-reprogrammed mature neutrophils. Interestingly, our results point to an infection-induced role for neutrophils during *T. brucei* infections that does not contribute to direct anti-parasite activity. In contrast, the expansion of enzymatically active neutrophils drives a detrimental mechanism related to an increased proteolytic MMP-8 and MMP-9 activity in the absence of sufficient TIMP-mediated inhibition. This, in turn, leads to the direct destructive

remodeling of the ECM by digesting the collagen and elastin components. In doing so, *T. brucei* parasites trigger the destruction of the tissue matrix needed to support the organization of B cell follicles and consequently hamper the generation of affinity-matured PCs. This infection-associated pathology can be dampened by the depletion of the expanding neutrophils and by improving the maintenance of the ECM and B cell follicle architecture. Consequently, neutrophil

**Fig. 6 | Crosstalk analysis between neutrophils and PCs.** Cell–cell interactions predicted by Nichenet. **a** Ligand–receptor interactions between plasma cells (PCs) (dark red) and neutrophils N1 (royal blue), N2 (dark green), N3 (violet), and N4 (steel blue). Genes representing a major BAFF signaling pathway are marked with an indicator (*). **b** Differential gene expression analysis between PCs of 14 dpi and naïve mice. Each dot represents a gene, with the *x*-axis showing the log2 fold change between conditions and the *y*-axis displaying the −log10 *p*-value from the differential expression analysis. Genes with statistically significant differential expression (*p*-value < 0.05) using the Wilcoxon rank sum test (adjusted using the Bonferroni correction) and a log2 fold change greater than 1 or less than −1, are colored in red, including *Tnfrsf17* (BCMA) and *Tnfrsf13b* (TACI). The expression levels of both receptors on PCs are represented in violin plots. Genes that did not meet these criteria are shown in green. **c** Heatmap of the prior interaction potential specific for

neutrophil ligands and PC receptors, highlighting the major pathway between BAFF (*Tnfsf13b*) and its receptors BCMA, TACI, BAFF-R (*Tnfrsf17, Tnfrsf13b,* and *Tnfrsf13c*), marked in pink. **d** UMAP feature plot representing the expression level of *Tnfsf13b* (BAFF) in spleen neutrophils and concentrations of BAFF in spleen extracellular homogenates and plasma of naïve mice and 14 dpi mice. Mean fluorescence intensity (MFI) of B-cell maturation antigen (BCMA) surface expression on PCs (B220$^{Int}$CD138$^+$) and non-PCs (B220$^+$CD138$^-$) from naïve and 14 dpi mice shows the reduction of receptor expression during infection. Data shows means ± S.D. of 3 mice from one of two validation experiments. An unpaired two-tailed Student's *t*-test was used for comparison, with ns indicating not significant for $p \geq 0.05$, and $p < 0.05$ considered as statistically significant. Source data are provided as a Source Data file.

depletion results in improved *T. brucei* parasitemia control and prolonged host survival.

## Methods

### Ethics statement
All experimental animal procedures were conducted according to EU directive 2010/63/EU and approved by the GUGC Institutional Animal Care and User Committee (IACUC), file numbers 2019-011, 2019.019.A, 2019-025, 2020-009, 2020-018, 2021-005, 2022-001, 2022-007, 2022-012, 2023-005, 2023-008, 2023-009, and 2023-012.

### Parasites and infection in mice
Female C57BL/6 wild type (6–8 weeks, Koatech, South Korea) were housed in IVCs with appropriate cage enrichment. Mice were infected by intraperitoneal (i.p.) injection of 5000 *Trypanosoma brucei brucei* AnTat 1.1E in 100 μL DPBS (Welgene, Gyeongsan-si, Gyeongsangbuk-do, South Korea). Parasitemia was assessed by a hemocytometer in blood collected from the tail vein (2.5 μl) at different time points during infection, diluted 1/200 in DPBS. For neutrophil depletion studies, mice were injected intraperitoneally (i.p.) with a solution containing 500 μg of the monoclonal antibody (mAb) anti-mouse Ly-6G/Ly-6C (Gr-1), clone RB6-8C5 (Biolegend, CA, USA) 2 days prior to parasite infection, followed by additional doses containing 100 μg of anti-Gr1 Ab on day 0, 2, 4, 6, 8, 10, and 12 post-infection. Control mice were treated using the same procedure but injected with an equivalent amount of an isotype control IgG2b antibody (clone RTK4530, Biolegend, CA, USA).

### Isolation of organs, peripheral blood cells, and plasma
Mouse spleens, femurs, and blood were isolated after the $CO_2$ euthanasia of the animals.

Single-cell suspensions were prepared by homogenizing spleens in 4 mL of Dulbecco's Modified Eagle Medium (DMEM) (Capricorn Scientific, Ebsdorfergrund, Germany) supplemented with 10% fetal bovine serum (FBS) (Atlas Biologicals, CO, USA) and 1% penicillin/streptomycin (Capricorn Scientific, Ebsdorfergrund, Germany), using gentleMACS™ Dissociator (Miltenyi Biotec, Bergisch Gladbach, Germany). After passing the homogenate through a 70 μm cell strainer (SPL Life Sciences, Gyeongi-do, Korea), cells were centrifuged at 314×*g* at 4 °C for 7 min. Cell pellets were resuspended in RBC lysis buffer (Biolegend, CA, USA) at 4 °C and incubated for 5 min. After washing (314×*g* at 4 °C for 7 min), cells were kept in FACSFlow Sheath Fluid (BD Biosciences, CA, USA) containing 0.05% FBS (Atlas Biologicals, CO, USA) on ice. Total live cells were counted by Trypan Blue (Sigma-Aldrich, Missouri, USA). Supernatants containing spleen extracellular homogenates were collected after centrifugation at 10,000×*g* for 10 min at 4 °C. Final supernatant was recuperated and frozen at −20 °C for further analysis.

Bone marrow cells were isolated by flushing femurs, using a 26-gauge needle and 1 mL ice-cold DMEM (Capricorn Scientific, Ebsdorfergrund, Germany) supplemented with 1% FBS (Atlas Biologicals, CO, USA) and 1% penicillin/streptomycin (Capricorn Scientific,

Ebsdorfergrund, Germany). Cell suspensions were centrifuged at 314×*g* at 4 °C for 7 min. Cell pellets were re-suspended in 3 mL FACSFlow Sheath Fluid (BD Biosciences, CA, USA) containing 0.05% FBS (Atlas Biologicals, CO, USA), and the total remaining live cells were counted using Trypan Blue (Sigma-Aldrich, Missouri, USA).

Blood was collected by cardiac puncture with a 1 mL syringe filled with 0.1 mL heparin (500 IU/mL). Samples were subsequently mixed with 20 mL RBC lysis buffer (Biolegend, CA, USA) at 4 °C for 15 min. After two wash steps with 20 mL ice-cold DMEM (centrifugation at 314×*g* at 4 °C for 7 min), blood cells were re-suspended in 1 mL of FACSFlow Sheath Fluid (BD Biosciences, CA, USA) containing 0.05% FBS (Atlas Biologicals, CO, USA) on ice. The total remaining live cells were counted by Trypan Blue (Sigma-Aldrich, Missouri, USA). For plasma preparation, heparinized blood samples were centrifuged at 2000×*g* at 4 °C for 10 min. Next, plasma was aliquoted into 3 Eppendorf tubes per mouse and stored at −20 °C for further analysis.

### Flow cytometry analysis
Surface cell staining was performed in FACSFlow Sheath Fluid (BD Biosciences, CA, USA) containing 0.05% FBS (Atlas Biologicals, CO, USA) using $10^5$ cells per sample. Nonspecific binding was blocked using 1/1000 diluted CD16/CD32 Fcγ III/II (Biolegend, CA, USA) for 20 min in the dark at 4 °C. Subsequently, cells were incubated with pre-determined optimal concentrations (Supplementary Table s1) of fluorochrome-conjugated antibodies and/or isotype controls for 30 min in the dark at 4 °C. After incubation cells were analyzed by flow cytometry analysis using a BD Accuri™ C6 Plus flow cytometer (BD Biosciences, CA, USA). Data was analyzed using BD Accuri™ C6 software, version 1.0 (BD Biosciences, CA, USA). Gating strategies are shown in Supplementary Figs. s1, s8, and s9.

For studying neutrophils apoptosis, caspase-3 activity measurement was performed using a CaspGLOW™ Fluorescein Active Caspase-3 Staining Kit (Biovision, Milpitas, CA, USA), following the manufacturer's instructions. Briefly, $10^6$ cells were incubated for 1 h with FITC-DEVD-FMK at 37 °C, 5% $CO_2$ in fresh pre-warmed DMEM medium (Capricorn Scientific, Ebsdorfergrund, Germany) in the dark. Next, cells were washed once with 2 mL FACSFlow Sheath Fluid (BD Biosciences, CA, USA) containing 0.05% FBS (Atlas Biologicals, CO, USA). Next, cells were centrifuged at 314*g* at 4 °C for 7 min before re-suspending in 100 μL of FACSFlow Sheath Fluid (BD Biosciences, CA, USA) containing 0.05% FBS (Atlas Biologicals, CO, USA) for subsequent staining with PE Ly6G, PE/Cy7 Ly6C and APC CD11b surface marker according to the same procedure described above. Data was analyzed by BD Accuri™ C6 Plus flow cytometer (BD Biosciences, CA, USA.

### Enzyme-linked immunosorbent assays (ELISA)
Concentrations of CXCL-12, CXCL-2, S100A8/S100A9, elastase, myeloperoxidase, lipocalin-2, lactoferrin, metalloproteinases MMP-8, MMP-9, MMP-25, TIMP-1, TIMP-2, and BAFF were measured in extracellular spleen homogenates and plasma samples, using the following kits according the manufacturer's protocols: Mouse

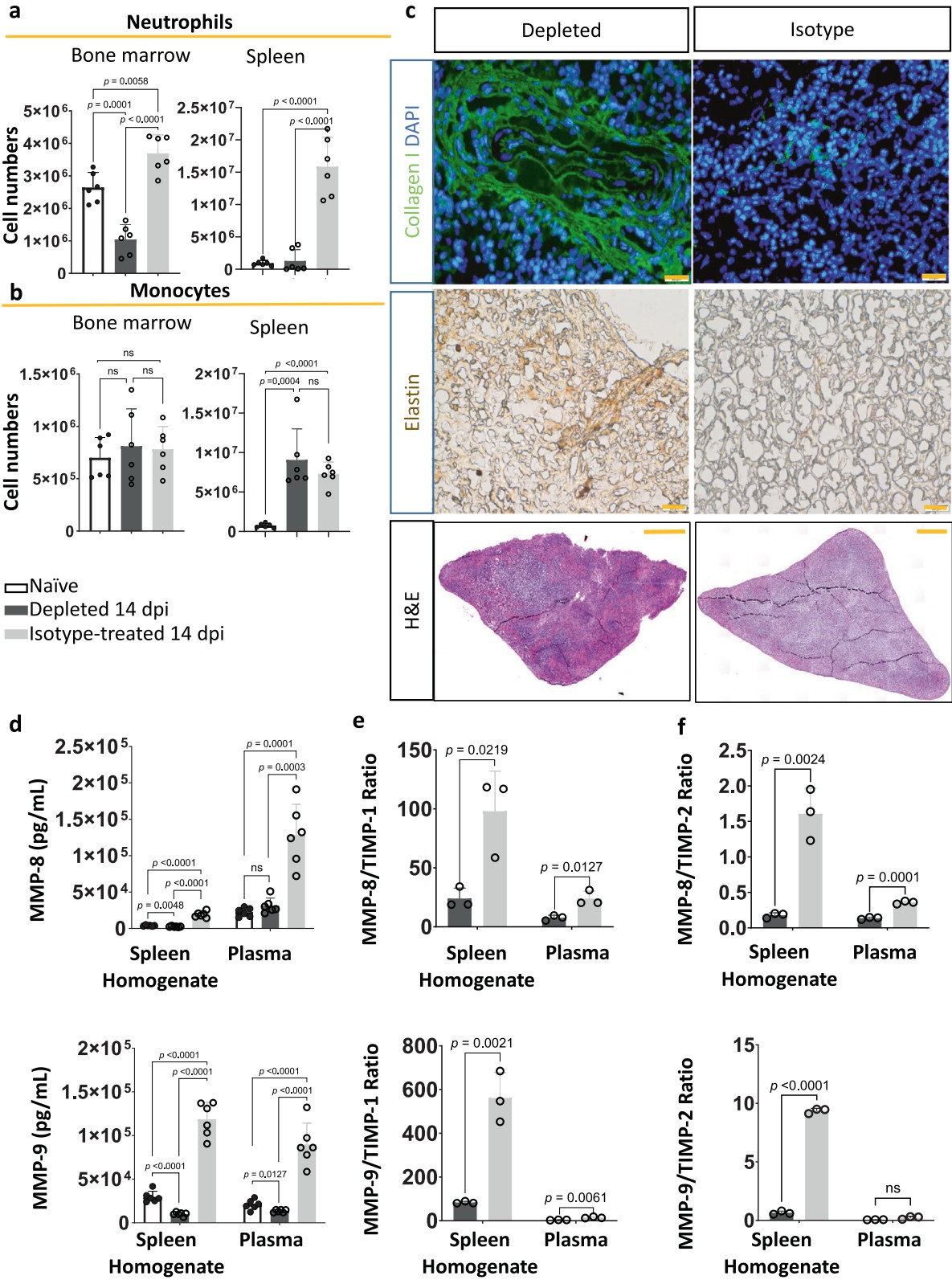

CXCL2/MIP-2 Quantikine ELISA Kit (R&D Systems, Minneapolis, MN, USA), Mouse CXCL12/SDF-1 alpha Quantikine ELISA Kit (R&D Systems, Minneapolis, MN, USA), Mouse S100A8/S100A9 Heterodimer DuoSet ELISA (R&D Systems, Minneapolis, MN, USA), Mouse Neutrophil Elastase ELISA Kit (Abcam, Cambridge, UK), Mouse Myeloperoxidase (MPO) ELISA Kit (Abcam, Cambridge, UK), Mouse Lipocalin-2/NGAL DuoSet® ELISA (R&D Systems, Minneapolis, MN, USA), Mouse Lactoferrin (LTF) ELISA Kit (Abbexa Ltd, Cambridge, UK), Mouse in vitro SimpleStep MMP-8 ELISA Kit (Abcam, Cambridge, UK), Mouse MMP-9 ELISA Kit (Abcam, Cambridge, UK), Mouse MMP-25 ELISA Kit (Abbexa Ltd, Cambridge, UK), Mouse TIMP-1 ELISA Kit (Abcam, Cambridge, UK), Mouse TIMP-2 SimpleStep ELISA Kit (Abcam, Cambridge, UK), and Mouse BAFF ELISA Kit (Abcam, Cambridge, UK).

**Fig. 7 | Neutrophil depletion prevents the destruction of spleen architecture and ECM damage. a** Neutrophil depletion from the spleen and bone marrow by anti-Gr1 mAb treatment, monitored by flow cytometry. Results are presented in bar graphs, showing total numbers of CD11b⁺CD177⁺ neutrophils present in naïve (white with black border), neutrophil-depleted/14dpi (dark gray), and isotype-treated/14 dpi (light gray) mice. **b** The impact of anti-Gr1 treatment on CD11b⁺ Ly6C⁺ monocyte numbers was assessed by flow cytometry in the same experimental groups. Data represent means ± S.D. collected from combined results of 6 mice from two of three separate experiments **c** Representative immunofluorescence images of frozen thin spleen sections stained with anti-collagen type I (green) and DAPI (blue) for nucleus staining. Immunohistochemistry was also used to stain elastin fibers with anti-elastin antibodies and a secondary antibody coupled to HRP (brown). In addition, thin spleen sections were stained with Hematoxylin and Eosin (H&E).

Scale bars, 20 μm (collagen and elastin) and 1 mm (H&E). For each condition, spleen sections from 3 individual mice per experimental group were stained, showing one representative result. **d** Concentrations of MMP-8 and MMP-9 in spleen extracellular homogenates and plasma were recorded in naïve, neutrophil-depleted/infected, and isotype-treated/infected mice using ELISA. **e, f** Bar graphs show the ratios between concentrations of MMP-8/TIMP-1 and MMP-9/TIMP-1, as well as MMP-8/TIMP-2 or MMP-9/TIMP-2 in spleen extracellular homogenates and plasma of neutrophil-depleted/14dpi (dark gray) compare to the isotype treated/14dpi (light gray) mice. Data represent means ± S.D from a combined 6 mice (two experiments) (**d**) or one of two validation experiments (3 mice per experimental group) (**e, f**). Unpaired Student's t-tests were used for comparison between experimental groups, with ns indicating not significant for $p \geq 0.05$, and $p < 0.05$ considered as statistically significant. Source data are provided as a Source Data file.

## Histopathology and immunofluorescence microscopy

Mouse spleens were isolated after $CO_2$ euthanasia of the animals and fixed overnight at 4 °C in neutral buffered 10% formalin solution (Sigma-Aldrich, Missouri, USA). Next, samples were soaked in 30% sucrose before being snap-frozen in FSC 22 Frozen Section Media (Leica Biosystems, IL, USA). Frozen 5 μm-thick sections were prepared at the Laboratory Animal Research Facility of the National Cancer Center, South Korea, and Duksung Women's University, Seoul, South Korea. For immunofluorescence staining, sections were fixed for 15 min in acetone:methanol (1:1 ratio). Slides were washed with wash buffer (PBS containing 0.1 BSA + 0.05%t Tween 20) before being blocked in PBS supplemented with 10% FBS (Atlas Biologicals, CO, USA) and 0.1% TWEEN 20 (Sigma-Aldrich, Missouri, USA) for 1 h at room temperature. Next, slides were incubated with primary antibodies overnight at 4 °C. After a wash buffer rinse, sections were incubated with secondary antibodies for 1 h at room temperature before counterstaining with DAPI (Sigma-Aldrich, Missouri, USA) diluted 1/1000 in DPBS for 5 min at room temperature. Finally, sections were washed three times in DPBS and mounted with VECTASHIELD® Antifade mounting medium (Vector Laboratories, Burlingame, CA, USA). Images were captured with an Olympus iX83 microscope integrated with an Olympus DP80 digital microscope camera and analyzed by Olympus cellSens® Dimension software V.1.18. Antibodies and their concentrations used for staining are listed in (Supplementary Table s2).

For IHC tissue sections elastin analysis, a horseradish peroxidase-diaminobenzidine (HRP-DAB) system staining kit (R&D Systems, Minneapolis, MN, USA) was used, following the manufacturer's protocol. Briefly, sections were blocked sequentially with Peroxidase Blocking Reagent for 5 min, Serum Blocking Reagent G 15 min, Avidin Blocking Reagent for 15 min then Biotin Blocking Reagent for 15 min, before 4 °C overnight incubation with the primary anti-elastin polyclonal antibody (Bioss Antibodies, Woburn, MA, USA). Next, sections were incubated with a biotinylated secondary antibody for 60 min at room temperature in the dark. Visualization of the specific binding is based on enzymatic conversion of the chromogenic substrate 3, 3⁰ diamino-benzidine (DAB) into a brown HRP precipitate. After counterstaining with hematoxylin (Abcam, Cambridge, UK), sections were examined using an Olympus iX83 microscope. For Hematoxylin and Eosin (H&E) staining, fixed sections were stained using the H&E staining kit (Abcam, Cambridge, UK) following the manufacturer's instructions.

## Single-cell RNA sequencing and data analysis

For the scRNA-seq characterization, splenocytes were obtained as described above from one naïve and one 14 dpi mouse after RBC lysis. Samples with viability >85% were used for sequencing using Nova-Seq6000 platform. Splenocytes per sample were partitioned using a 10X Genomic Chromium Controller. Libraries were constructed using the Chromium Single Cell 3′ v3 reagent kit following the manufacturer's instructions (10× Genomics, Pleasanton, CA). Raw sequencing reads were processed using the Cell Ranger version 5.0.0 pipeline.

Briefly, the Illumina sequencer's base call files (BCLs) for each flow cell directory were demultiplexed into FASTQ files using the *cellranger mkfastq* command. The *cellranger count* command was used to align the sequencing reads (fastq) to the *Mus musculus* reference genome (mm10) and quantify the expression of transcripts in each cell. This approach resulted in the generation of a gene expression matrix for each sample, which records the number of UMIs for each gene associated with each cell barcode. After the alignment, a UMI matrix per gene per sample was generated for every sample (naïve and 14 dpi). Initially, 9991 splenocytes (55,468 mean reads/cell and 1734 median genes/cell) for naïve and 10,619 splenocytes (49,601 mean reads/cell and 1,655 median genes/cell) for 14 dpi were captured. Next, the *cellranger count* outputs were loaded in SoupX using the *Read10X* function. The ambient background RNAs contamination was removed from the feature-barcode matrices obtained from CellRanger with the default SoupX (version 1.5.2, Wellcome Sanger Institute, Hinxton, Cambridgeshire, UK) workflow (autoEstCounts and adjustCounts)[70] in RStudio (version 1.3.1093, RStudio, Inc., Boston, Massachusetts, USA), before being processed using Seurat (version 4.0.5, New York Genome Center, New York City, New York, USA), into two separated Seurat objects for the naïve and 14 dpi sample[71,72]. Quality control filtering was applied to remove any cell where there were either (i) fewer than 200 unique features, (ii) greater than 4000 or 5000 unique features for naïve and 14 dpi samples, respectively, or (iii) greater than 10% mitochondrial genes. Before normalization, genes Gm42418 and AY036118, causing technical background noises[73], were removed from both datasets. All numerical data outputs are compiled in Table s3. The Seurat standard integration approach was applied to combine datasets from naïve and 14 dpi samples. The Seurat R package was used to normalize datasets using the global-scaling normalization method *LogNormalize* in which the gene counts for each cell were divided by the total gene counts for the cell and multiplied by a scale factor (10,000 by default). Next, the *FindVariableFeatures* command was used to select 2000 highly variable genes to compute a principal component analysis (PCA), of which the output was used for non-linear dimensionality reduction using the Uniform Manifold Approximation and Projection (UMAP) method. The integrated datasets were then run through the standard workflow for visualization and clustering, including *ScaleData* to eliminate cell–cell variation in gene expression driven by batch effects, *RunPCA, RunUMAP* (30 dimensions), followed by *FindNeighbors* (30 dimensions, reduction = "pca") and *FindClusters* (resolution = 0.5). To identify different cell populations present in the dataset, first SingleR (v1.4.1) was employed for automatic cell annotation[74] by mean of the ImmGen database, followed by verification using canonical gene markers collected from literature[5,6,35–39]. Cells annotated as neutrophils by SingleR were extracted from the combined dataset using the Seurat *Subset* function to create a new Seurat object[75]. The latter was subjected to another round of dimensionality reduction and clustering analysis by *FindNeighbors* (4 dimensions, reduction = "pca") and *FindClusters* (resolution = 0.1), resulting in 4 neutrophil subclusters. To obtain DEG among annotated subclusters

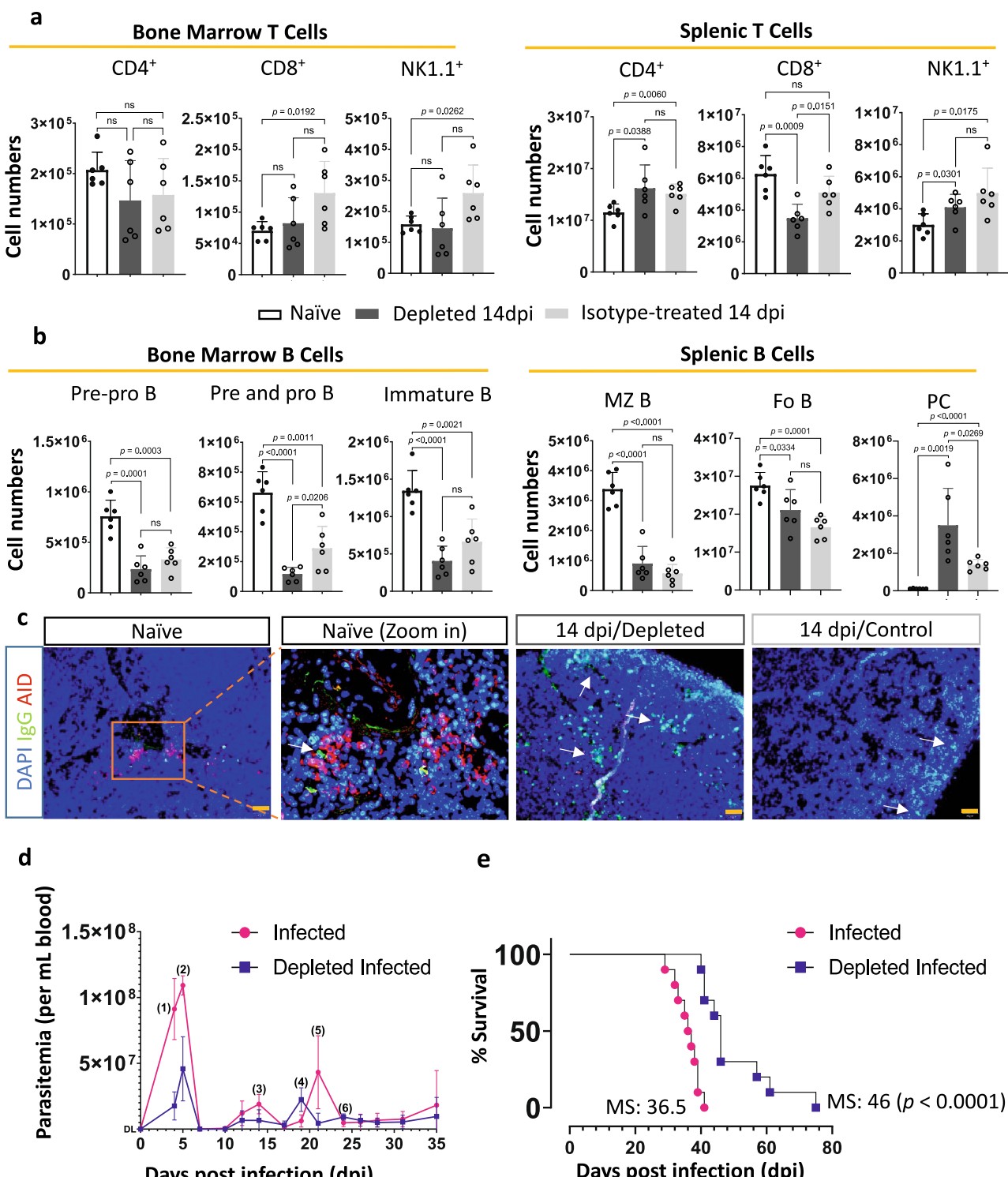

**Fig. 8 | Neutrophil depletion induces PC generation and improves parasitemia control and survival.** Samples of naïve, neutrophil-depleted/14dpi and isotype treated/14dpi mice were analyzed by flow cytometry (gating strategies in Figs. s8 and s9). **a** Total number of CD4+, CD8+, and NK1.1+ NKT/NK cells. **b** Total number of B cells including B220+IgM−CD19− Pre-Pro B cells, B220+IgM−CD19+ Pro and Pre B cells, B220+CD93+ Immature B cells, B220+Cd1dHigh MZBs, B220+Cd1dLow FOBs cells and B220IntCD138+ PCs. For **a**, **b**, Data represent means ± S.D collected from combined results of 6 mice from two of three separate experiments using unpaired two-tailed Student's t-tests for analysis, with p ≥ 0.05 considered as not significant (ns), and p < 0.05 considered as statistically significant. **c** Representative immuno-fluorescence images of thin frozen spleen sections of all experimental groups stained with anti-IgG antibody (green), anti-AID antibody (red), and DAPI (nuclear staining−blue). Scale bar = 50 μm. White arrows indicate IgG+ B cells. Spleen sections from three individual mice per experimental group were stained, showing one representative result. **d** Parasitemia in neutrophil-depleted mice (blue) and control-infected mice (dark pink). Mice received one injection of anti-Gr-1 antibody or isotype control two days before the *T. b brucei* AnTat 1.1 trypanosomes challenge, followed by subsequent antibody treatment every two days till 12 dpi. Pairwise comparison was conducted between the two groups using unpaired two-tailed Student's t-tests, with p = (1)0.0009 (2)0.0031 (3)0.0347 (4)0.0115 (5)0.0329 (6) 0.00306 considered as statistically significant. **e** Survival of depleted (blue) and control (dark pink) mice. For **d**, **e**, results represent combined data from 2 individual experiments, using ten mice per experimental group in total, using a Mann Whitney U test for analysis of parasitemia and a Mantel−Cox test for survival curves (MS = median survival). Source data are provided as a Source Data file.

or between 14 dpi and naïve datasets, the Wilcoxon rank sum test was performed with the *FindAllMarkers or FindMarkers* function (*test.use*='Wilcox', *min.pct* = 0.25) of Seurat. The adjustment of *p*-values was performed using a Bonferroni correction based on the total number of genes in the dataset. DEGs were ranked using $Log_2$ fold change and adjusted *p*-value. Genes with *p*-values > 0.05 were omitted. A pseudo-time plot was generated with Monocle version 3 to infer the potential lineage differentiation trajectory[76]. The *as.cell_data_set* function was used to build a Monocle object from the normalized counts of cells present in the four neutrophils clusters based on Seurat analysis. The cells were ordered in pseudotime along a trajectory using *reduction_dimension* with the UMAP method and *order_cells* functions. The N1 subcluster was selected as the starting point (initial stage). Module scoring for certain biological functions was calculated using *AddModuleScore* function. The functional scores were defined as the average scaled expression of corresponding genes. Signature genes were chosen based on previously published data[35–37] or selected DEGs between 14 dpi and naïve datasets. The data were then represented in violin plots for the module scores as well as heatmaps using genes included in the module. An unpaired Student *t*-test was used to compare the mean module scores between naïve and 14dpi datasets. Multibar heatmaps were constructed from the annotated Seurat object using DoMultiBarHeatmap package (https://github.com/elliefewings/DoMultiBarHeatmap). These heatmaps were constructed based on DEGs or on scaled gene expression data using previously published gene markers[35–37] related to various cellular processes.

BM cells from femurs of three naïve and three 14 dpi mice were analyzed using scRNA-seq, similar to the methodology outlined for splenocytes. The Cell Ranger pipeline (version 7.0.1) processed 8579 and 8020 BM cells for naïve and 14 dpi conditions, respectively. Data were cleaned with the SoupX (version 1.6.2) and Seurat packages (version 4.3.0), with genes causing technical noise (Gm42418 and AY036118) removed[70–73]. Quality control for BM data was stringent, filtering out cells with fewer than 100 unique features, more than 6000 unique features, or greater than 10% mitochondrial genes. This ensured the exclusion of low-quality cells. Subsequently, the naïve and 14 dpi datasets were merged and normalized. The integrated dataset was further refined with Harmony (version 0.1.1)[77] to remove batch effects and cell–cell variation. Dimension reduction and clustering were carried out using RunPCA, RunUMAP (26 dimensions), Find-Neighbors (26 dimensions, reduction = "harmony"), and FindClusters (resolution = 0.5) functions. Cell type annotation was achieved with the SingleR package (version 2.0.0), leading to the identification of neutrophil gene-expressing cells using previously published markers[35–37,40]. This cluster was further divided into four distinct neutrophil subclusters. DEGs among the subclusters and between the naïve and 14 dpi datasets were identified using the Wilcoxon rank sum test, and the top DEGs were visualized with the DoMultiBarHeatmap package (version 0.1.0).

To assess potential cell-cell communication, the NicheNet framework was used[47], focusing on interactions between neutrophils ligands and corresponding receptors on PCs or FoBs, with default "mouse" parameters comparing gene expression between experimental groups ("14dpi" versus "naive"). NicheNet integrated data were ranked based on their likelihood in relation to observed experimental expression changes. Highest-ranked interactions were selected for further analysis and visualization in heatmaps, feature plots, and violin plots.

### Graphs and statistical analyses
Unless otherwise indicated, statistical analysis was done using an unpaired Student's *t*-test, comparing data from infected mice vs. naïve values. The values shown in each figure represent means ± S.D. Values of $p \leq 0.05$ are considered statistically significant. All statistical analyses and graphics were made using GraphPad Prism V.8.3 (GraphPad

Software Inc. San Diego, CA, USA). A Mann–Whitney *U* test was used for statistical analysis of parasitemia. Survival curves were analyzed using the Mantel–Cox statistic, using median survival data (MS) for comparison.

### Reporting summary
Further information on research design is available in the Nature Portfolio Reporting Summary linked to this article.

## Data availability
The generated scRNAseq data obtained from the spleen and bone marrow in this study have been deposited in the Omnibus Gene Expression database under accession codes GSE222784 for spleen (including Naïve–GSM6932272 and 14 dpi–GSM6932273) and GSE234000 for BM. In addition, the Naïve control data can also be extracted as a subset of the ArrayExpress E-MTAB-10174 deposition. Source data are provided in this paper.

## Code availability
The codes generated during this study have been deposited in the Github repository and can be accessed at Zenodo.org as following links: https://doi.org/10.5281/zenodo.8232243 (for spleen) and https://doi.org/10.5281/zenodo.8232730 (for bone marrow).

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

## Acknowledgements

We thank members of the Laboratory Animal Research Facility at the National Cancer Center and HyunKyung Lim from Duksung Women's University, South Korea, for their technical support related to tissue processing for immunohistochemistry. We would like to express our gratitude towards Prof. Jonathan Ozelton for his support in reviewing and editing the paper. We extend our gratitude to Dr. Hang Thi Thu Nguyen for her valuable assistance in preparing the samples for scRNAseq, as well as her input during the discussions on cell annotation. This work was supported by Ghent University Global Campus core funding, UGent BOF grant number BOF.STG.2018.0009.01/01N01518, FWO grant number G013518N, and the Vrije Universiteit Brussel, grant number SRP63.

## Author contributions

Conceptualization: M.R. and S.M. Investigation: H.T.T.P., M.R., S.M., B.C., B.B. and J.J. Bioinformatic: H.T.T.P. and B.C. Data analysis and interpretation: M.R., H.T.T.P., S.M. and B.C. Writing: M.R., S.M., H.T.T.P. and B.C. Critical revision: M.R., S.M. and J.J. Visualization: H.T.T.P., B.C. and S.M. Funding acquisition: M.R. and S.M.

## Competing interests

The authors declare no competing interests.
