## [Peer Review File · Nature Communications]

Neutrophil metalloproteinase driven spleen damage hampers infection control of trypanosomiasisREVIEWER COMMENTS

Reviewer #1 (Remarks to the Author):

The authors have performed an interesting analysis of the role of neutrophils and their products - MMPs in causing pathology in Trypanosome infections in an animal model. Their work elucidates an important description of the four subsets of neutrophils that arise in response to experimental infection. Their work also highlights the important role of neutrophils in this response by using neutrophil depletion experiments.

Concerns to be addressed:

A. The first part of the study involves the description of the neutrophilic response in the spleen of infected mice and the difference from naive mice. This leads to the following questions:

1. How specific is the response to Trypanosomes? Can any pathogen cause the same response? How can the authors confirm specificity as currently designed?
2. The authors do not examine the kinetics of the response. Only a single time point (14 days post-infection) is chosen and all data obtained. This considerably weakens the analysis. What is the normal kinetics of Trypanosoma infection in this setup. What were longitudinal studies not performed?
3. It would have been interesting to examine the neutrophil profile in bone marrow and blood in addition to spleen at this time point even if neutrophilic was not observed.

B. The second part of the study describes the function of neutrophils as assessed by RNA and protein expression in spleen and plasma.

1. Again, without a clear kinetics of neutrophil function being examined, it is hard to conclude if any of the other markers had no role in this infection model.
2. Why were other MMPs not measured in spleen or plasma?
3. What were the levels of TIMPs in spleen or plasma?

C. The final part is the functional assessment following neutrophil depletion.

1. The data clearly show that neutrophils are important in survival and plasma cell regulation etc but whether any of this is linked to MMPs is not clear.
2. What experiments would the authors perform to delineate the role of neutrophil derived MMPs in their model? Why was this not performed?
3. What happens to other neutrophil functions in these depleted animals?

D. The authors should discuss this data in the context of human infection and add any relevant data from human samples to add value to their findings.

Reviewer #2 (Remarks to the Author):

In this manuscript, Pham et al describe for the first time the impact of *T. brucei* infection on splenic neutrophil populations. Through a series of elegant *in silico* and *in vivo* analyses, the authors propose a model whereby neutrophils impact the spleen architecture impairing adequate adaptive responses against the parasite. There is an incredible wealth of information coming out of this paper and I would like to commend the authors for such a clear and concise story. My comments below are only aimed at increasing the clarity of the manuscript and figures, and I hope the authors find them constructive. One major point to raise is perhaps the overall lack of a deeper characterisation of the various splenic populations identified by single cell transcriptomics. On a similar note, several prediction tools are available to generate hypotheses as to how these various immune cells might be interacting with one another. These analyses would certainly elevate the manuscript even further.

Introduction

Extremely clear and well balance. I commend the authors for clearly summarising such a complex field in such a concise and clear way.

- Line 48 – Reference 6 is missing.

Methods

- The authors should consider depositing their scripts and processed files (e.g., rds files) in a public repository like Zenodo to facilitate knowledge exchange.
- The various QC outputs should be included somewhere in the supplementary figures to increase transparency.

Results

- Did the authors explore the effect of infection on other splenic populations at the single cell level? For instance, it would be interesting to include a characterisation of the B cell compartment within the spleen, in which one could hypothesise an increase expression of apoptotic genes?
- Have the authors considered performing cell-cell interaction analysis to predict which ligand-receptor pairs might be driving the interaction between neutrophils and other effector populations in the spleen? One obvious choice would be neutrophil-B cells to predict which interactions might be driving the depletions observed in the spleen and previously reported in this infection system.
- Although I appreciate the manuscript focusses on neutrophils, and this is well justified, I feel like a deeper characterisation of the other splenocyte populations is required as this would be an incredible resource to the community. Some more detailed analysis such of the various innate and adaptive cell types captured in this experiment would be required.
- Line 106: The authors state that infection does not impact the levels of neutrophils in blood and BM, but according to figure 1b there is a significant difference between groups, however small. Can the authors comment?
- Regarding figure 1C, the resolution used for dimensionality reduction and UMAP projection does not seem to be included in the methods. Also, it is unclear how the various basic scRNAseq parameters compare before and after filtering. This information should be made available as a supplementary file prior to acceptance of this manuscript.
- Regarding figure 1C, it would be more informative to relabel the different clusters as the proposed cell IDs (for instance, based on the SingleR analysis) to increase the clarity of what the various cell clusters are. This is provided in figure S5a, but it needs to be made clearer in 1C, as it looks as though all these clusters as neutrophil-related.
- Figure 2A, the word "UMAP_" is missing a number.
- In figure 2, can the authors include the maker genes for each of these populations (or at least the ones referenced in the text – i.e., Cebpe, Ccl6, Il1b) as feature or violin plots? It would help with clarity.
- In figure 2C, it is not clear whether the N1 population is not present in the naïve samples, or whether it was excluded? Can the authors comment this on the text? Is it possible that N1 appears as a consequence of emergency granulopoiesis?
- Line 135 - 148: Can the authors reference the relevant figures showing the genes listed in the text? It is not immediately obvious.
- In figure 3a (and throughout) the authors could include an asterisk or arrowhead on the genes discussed in the text to highlight them. This is to facilitate the reading and interpretation of the figures.
- Stats in figure 3e are missing.
- Is there a corresponding quantification of the results presented in figure 3f?
- Line 204: "none of the 7 markers used, shows" needs rewording
- Figure 4e – can the authors comment on the specificity of circulating levels of LCN2 and LTF as a proxy for neutrophil function?
- Line 259: has this been observed in other conditions? If so, probably worth referencing here.
- Figure 6b is missing stats on BM results.
- Line 283: can the authors include this datum somewhere in the figure? I think it is important.

Discussion

- As mentioned above in the results sections, I wonder whether the N1 splenic population can be the result of emergency granulopoiesis? I appreciate this would happen in the BM, but it might be

a signature derived from this process. Can the authors comment about this possibility in the discussion?

- How can the authors reconcile the changed transcriptional landscape of mature neutrophils, as stated in line 383, with their so far known lack of transcriptional activity?

Reviewer #3 (Remarks to the Author):

In this elegant study, authors have used the single cell RNA-seq in conjunction with the current knowledge to show the impact of neutrophils expansion in context of *T. brucei* infection and spleen pathology. The story of the manuscript is tightly and elegantly presented.

My main concern is the number of mice used and if data are reproducible considering few mice are used. In majority of figures authors have added a statement that presented data is from 3 mice/experiment with 1-2 experiments. Three mice are not sufficient for power or significance analysis and draw conclusions.

Considering the pathology of Tb infection is in brain, it would be nice to see the effects of splenic depletion of neutrophils on brain pathology. Does it change?

Reviewers' comments and answers:

Specific answers to reviewers:

Reviewer 1:

The authors have performed an interesting analysis of the role of neutrophils and their products - MMPs in causing pathology in Trypanosome infections in an animal model. Their work elucidates an important description of the four subsets of neutrophils that arise in response to experimental infection. Their work also highlights the important role of neutrophils in this response by using neutrophil depletion experiments.

We thank the reviewer for the positive feedback at the start of this comment section.

Concerns to be addressed:

A. The first part of the study involves the description of the neutrophilic response in the spleen of infected mice and the difference from naive mice. This leads to the following questions:

1. How specific is the response to Trypanosomes? Can any pathogen cause the same response? How can the authors confirm specificity as currently designed?

This study is the first in its kind to show the detrimental effect of infection-associated neutrophil activation at the level of spleen architecture, resulting in a negative effect on parasitemia control of eukaryotic parasites. As we have outlined in the text, this mode of immune activation aids the parasite in partially escaping immune destruction. Whether or not there are other parasites that follow a similar path to escape immune destruction is currently unknown, as no similar observations have been reported before. Hence, our results can open the door for others, to study parasite-host interactions in light of these new findings. With respect to viral and bacterial infections, we have indicated the very recent finding, that reprogramming of neutrophil activity has been proposed as a key feature in the fight against COVID-19 associated pathology (discussion line 408). In line with this, a 'pathological' role for activated neutrophils has been described in fecal-induced sepsis as well as in pancreatic carcinoma models (line 354). Hence, these reports suggest that the in-depth neutrophil characterization provided in our study will benefit the future understanding of neutrophil biology in a wider scope, and not just in the domain of eukaryotic parasite (or trypanosomiasis) research.

2. The authors do not examine the kinetics of the response. Only a single time point (14 days post-infection) is chosen and all data obtained. This considerably weakens the analysis. What is the normal kinetics of Trypanosoma infection in this setup? What were longitudinal studies not performed?

With full respect for the effort this reviewer has taken to analyze our manuscript, we would like to point towards our study that preceded this paper (ref 19). Here we did a timeline follow-up of neutrophil accumulation in function of parasitemia, over a full infection period of 28 days. This showed that at 14 dpi, there is a maximal accumulation of neutrophils in the spleen, after which (17dpi) numbers slightly decline again. As the 14 dpi timepoint coincides with the onset of severe immune destruction, loss of B cell activity and destruction of the secondary follicular immune organ architecture (proper references are provided throughout the text), this point was chosen for the in-depth analysis presented here. To highlight this aspect of the study even better, we have rephrased the first lines of the result section (line 98 - 101), and the original finding has been mentioned in both introduction and discussion sections.

3. It would be interesting to examine the neutrophil profile in bone marrow and blood in addition to spleen at this time point even if neutrophilic was not observed.

While the paper was under review, we initiated the study of the bone marrow (BM) neutrophil population in the same *T. b. brucei* AnTat 1.1 setting, as the one used for this paper (14 dpi, triplicate biological samples for both infected and naïve groups). A study on blood transcriptomics is planned for the future. Hence, while

an editorial comment indicated that no additional transcriptomic data was required for resubmission, a full BM data set has been added to this study (Fig. 2e and new Suppl. Fig. s4). Important to flag is that this new data confirms the initial flow cytometry data (Fig. 1) indicating that only minimal changes in the BM neutrophil compartment take place during infection (in contrast to the spleen, see result text lines 123-129, discussion lines 386-395). As part of this publication, all new scRNAseq data obtained in the new BM study will be made available to the scientific community through deposition into the Omnibus Gene Expression database. This will aid the community in future data mining.

B. The second part of the study describes the function of neutrophils as assessed by RNA and protein expression in spleen and plasma.

1. Again, without a clear kinetics of neutrophil function being examined, it is hard to conclude if any of the other markers had no role in this infection model.

This study does indeed not exclude a role for other cells (besides neutrophils) in the pathology of trypanosomiasis. In fact, multiple studies on B cells, T cells, NK cells and macrophages/monocytes have shown that the pathology of *T. b. brucei* infection is complex and involves a systemic inflammatory modulation of various immune compartments. However, linking neutrophil activation with B cell dysfunction, and the subsequent detrimental effect on the capacity of the immune system to control infection, has never been shown before. As outlined above, the rationale for our focus on 14 dpi is based on our previously reported work.

2. Why were other MMPs not measured in spleen or plasma?

When analyzing the *Mmp* expression data, *Mmp8* and *Mmp9* appear as Top10 differentially expressed genes (DEGs) for subpopulation clustering (Suppl. Fig. s3). These genes are considered as key neutrophil markers, hence their inclusion in the original paper. However, as *Mmp25* can be found in the DEG list as well, we have now included both expression and protein data for this molecule (Fig. 5). MMP-25 levels (ELISA) were found to be extremely low compared to MMP-8 and MMP-9 values. We have also added the *Mmp25* feature plot, corroborating the minor expression compared to *Mmp8* and *Mmp9*. Finally, we have provided gene expression data for all three *Mmps* (new violin plots in Fig. 5) showing that the main pathological issue with *Mmp* expression can be linked to the exorbitant expansion of the N2-N4 neutrophil populations during infection (N2 and N3 are virtually absent in healthy naïve mice). This finding has been added to the text (lines 235-246).

3. What were the levels of TIMPs in spleen or plasma?

TIMP data were previously included as part of the MMP/TIMP ratio figures, but have now been added 'as measured' to Supplementary Fig. 7, as well as visual representation of the gene expression levels in the various neutrophil populations (data that was previously included in the Supplementary Table s3). To enhance clarity, we have adjusted the corresponding text (lines 301-308). In the main figure set, we have included the separate data for MMP-8 and MMP-9, as well as the MMP/TIMP ratios, as the latter key for understanding the biological role of these proteases.

C. The final part is the functional assessment following neutrophil depletion.

1. The data clearly show that neutrophils are important in survival and plasma cell regulation etc but whether any of the this is linked to MMPs is not clear.

First, we have now included levels of MMP-8 and MMP-9, showing their reduction upon neutrophil depletion (Fig. 7d, text lines 297-299). In our view, the neutrophil depletion approach has shown that infection induced neutrophil activation is a pivotal event in the induction of follicular architecture destruction (Fig. 7c), which in turn is known to be crucial for B cell affinity maturation and PC differentiation. MMPs were shown to be a crucial actor in this destructive event. We have however been very careful not to suggest that these are 'the only' drivers of pathology, but have indicated that based on the DEG analysis, these are the only major effector proteases that can be directly associated to tissue matrix destruction through their enzymatic activity. Depleting infection-induced neutrophils shows a direct link to MMP reduction, and reduction of ECM damage (Fig. 7). This is amply discussed (412-440). The detrimental role of MMPs during human trypanosome infections has moreover been shown by others (ref 27, 28 Line 422).

2. What experiments would the authors perform to delineate the role of neutrophil derived MMPs in their model? Why was this not performed?

Our *in vivo* neutrophil depletion studies show a relation between increased MMP expression by the infection-induced neutrophil activation and the infection associated destruction of the spleen architecture, which is coupled to B cell dysfunction and overall breakdown of the host adaptive immune response. To our knowledge, at this stage, there are no better *in vivo* tools available to assess whether this correlation can be explained by a direct causal relation, or would involve any intermediate step. The only way to show direct causality would be the use of neutrophil cell-specific KO mice for MMP8, MMP9 and MMP25 separately, as well as the combination of cell-specific 'double KOs', and ultimately the neutrophil cell-specific MMP8/9/25 'triple' KO mouse. Unfortunately, none of these mice are available.

At first sight, an alternative approach could be the use of neutrophil deficient mice (MRP8-Cre Mcl-1^{fl/fl}, hMRP8-ATTAC, G-CSFR^{-/-}, Cxcr2^{-/-}, Gfi-1^{-/-}, LysM-Cre Mcl-1^{fl/fl} or Foxo3a^{-/-} or so-called Genista mice), but all these suffer from various biological artifacts (including impaired cell specificity, off-target effects, high mortality and poor breeding capacity). Most importantly however, their use would not prove the specific MMP neutrophil axis (only overall neutrophil involvement). Also, neutralizing MMPs by a systemic approach would make it impossible to exclude the role of other organs (such as the liver) or cells in the MMP contribution to the observed shortened overall host survival or deteriorated parasitemia control. Hence, in our view, the approach of measuring specific MMP levels in spleen (and plasma), combined with our spleen transcriptomic data and neutralization approach, is the best way to support the idea that infection-induced spleen neutrophil activation and MMP production are important players in *T. brucei* associated pathology. Finally, we would like to emphasize that our study shows the heterogeneity of infection-induced neutrophil subpopulations which should be 'warning' for choosing one single gene target when designing a future neutrophil KO mouse model approach for infectious disease studies.

3. What happens to other neutrophil functions in these depleted animals?

We are intrigued by this rationale for the question, as it is not clear to us which neutrophil functions the reviewer is alluding to, in 'neutrophil depleted mice'. As outlined in the study, we only focused on trypanosomiasis-associated pathology related functions.

D. The authors should discuss this data in the context of human infection and add any relevant data from human samples to add value to their findings.

As mentioned above, and in the discussion section, *Mmp9* has been found in human PMBCs as a marker for severe disease, with MMP-9 itself being increasingly present in CSF when parasites pass the BBB and infiltrate the central nervous system (lines 419-423). As for a functional human spleen neutrophil analysis, this would require a multiple biopsy approach, which has never been attempted as far as we know in HAT patients. It

would involve significant ethical considerations. With respect to breakdown of the B cell adaptive immune system in HAT, there is data available showing that sleeping sickness patients suffer from a reduced specific antibody induction capacity, similar to what we have observed in the *T. brucei* mouse model used in this manuscript.

Reviewer 2:

In this manuscript, Pham et al describe for the first time the impact of *T. brucei* infection on splenic neutrophil populations. Through a series of elegant *in silico* and *in vivo* analyses, the authors propose a model whereby neutrophils impact the spleen architecture impairing adequate adaptive responses against the parasite. There is an incredible wealth of information coming out of this paper and I would like to commend the authors for such a clear and concise story.

We would like to start by thanking this reviewer for the level of positivity in the work assessment, and the way the work is commended. We also want to thank the reviewer for the very thorough reading and the many detailed questions below, which have allowed us to further improve the work and its impact.

My comments below are only aimed at increasing the clarity of the manuscript and figures, and I hope the authors find them constructive. One major point to raise is perhaps the overall lack of a deeper characterisation of the various splenic populations identified by single cell transcriptomics. On a similar note, several prediction tools are available to generate hypotheses as to how these various immune cells might be interacting with one another. These analyses would certainly elevate the manuscript even further.

As outlined above, we thank the reviewer for these general comments. We have now included a full detailed spleen cell annotation (adjusted Fig. 1, and Supplementary Fig s2, corresponding text lines 106-110), and as suggested, have used several prediction tools to make an even 'deeper' analysis of the interaction between infection-induced neutrophils and B cells, in the model of *T. b. brucei* AnTat 1.1. (New Fig. 6, and Suppl. Fig. s6, entire new text section lines 254-288). By doing so, we hope that we can satisfy the curiosity of the reviewer even further.

The current analysis predicts a major interaction between infection-induced neutrophils and plasma cells (PCs) involving the BAFF signaling pathway. With PCs downregulating both survival and differentiation receptors, this would correlate with the decreased capacity of the host to produce high-affinity specific anti-trypanosome antibodies. Interestingly, this new cell-cell interaction analysis further corroborates the overall detrimental role of neutrophils on parasitemia control, and hence links directly to the final Fig. 8 (previous Fig. 7) in which we have shown that neutrophil depletion actually improves parasitemia control and prolongs the survival of the host.

The Supplementary Figure s6 has mirrored the analysis for neutrophil-FoB interactions, as these cells are the major precursors for PCs in a mature immune system. The current approach does not allow for an analysis of interactions of neutrophils with other major spleen B cell populations such as MZBs, as these cells are virtually absent at 14 dpi. Likewise, a virtual analysis of neutrophils and GC interactions has been omitted for a similar reason, as the current data (and our previously published reports) show that at 14 dpi a near-complete destruction of the GC architecture takes place, not allowing for proper cell-cell interactions. Interestingly however, both the PC- and FoB-neutrophil interactome analysis are corroborating the major pathological B cell observations that were predicted in the original version of this manuscript. A discussion is included in lines 452-462.

A final comment relating the analysis of 'all' other scRNA-seq spleen immune cell data, is that while interesting, there is no 'ideal single timepoint' to do this. In fact, we are convinced that a whole battery of scRNA-seq experiments is needed, complemented with other approaches, targeting different timepoints at

which different cell-specific pathologies occur in different organs and body locations. For example, MZB cells vanish right at the first peak of infection, and as indicated above, the 14dpi timepoint is not suited for analyzing these cells. Likewise, while FoBs are still present at 14 dpi, their numbers start to decline around week 3 post infection. In line with this thinking, there are surely biological events in the liver, brain and muscle or fat tissue (and even peritoneum) that are crucial for understanding the overall pathology of *T. brucei* induced pathology. Hence, we consider the data that is provided in this manuscript as uniquely valuable to progress our understanding of HAT and AAT, but at the same time humbly realize that many more pieces of the puzzle remain to be added.

Introduction

Extremely clear and well balance. I commend the authors for clearly summarising such a complex field in such a concise and clear way.

We thank the reviewer for this positive comment

- Line 48 – Reference 6 is missing.

We very sincerely apologize for the fact that that '6' dropped out of the manuscript during one of the many re-writes that were needed to get the word-count of the manuscript within the journal wordcount limit. This has been corrected (Line 49). We are particularly sorry for this mistake as this reference was the first to give a holistic view on transcriptomic profiling of the brain pathology of experimental trypanosomiasis, and as such has to be considered as a milestone in the field. That is why the reference was included in the reference list, and was supposed to have appear very early-on in the manuscript. This problem has been fixed now.

Methods

- The authors should consider depositing their scripts and processed files (e.g., rds files) in a public repository like Zenodo to facilitate knowledge exchange.

We have deposited all data in the Omnibus Gene Expression database and uploaded the codes used in the analysis to our Github repository, as mentioned in the 'Data Availability' and 'Code Availability' sections. In addition, we have now also uploaded all BM new scRNA-seq data. Access was however put on-hold for a period of 6 months (during the initial submission period). Accession are: GSE222784 and GSE234000.

- The various QC outputs should be included somewhere in the supplementary figures to increase transparency.

We have added a very comprehensive supplementary data set (SP and BM) outlining all QC and data analysis procedures (Supplementary Table s3).

Results

- Did the authors explore the effect of infection on other splenic populations at the single cell level? For instance, it would be interesting to include a characterisation of the B cell compartment within the spleen, in which one could hypothesise an increase expression of apoptotic genes?

We have now included a full annotation of spleen (both Naïve and 14 dpi) in the adjusted Fig. 1 (and Suppl. s2), and have included (as indicated above) a completely new Fig. 6 (and Suppl. Fig. s6). The complementary new text paragraph (254-288) explores the impact of infection-associated neutrophil activation in particular on the PC and FoB cell populations. Due to the word limit imposed by the journal, some of the other overall data description had to be shortened because of this, without altering any of the conclusions of the first version of the paper. The main finding, as already mentioned above, is the downregulation of B cell survival

receptor gene expression for BCMA, TACI and BAFF-R during infection. For BCMA we were able to confirm the result by flow cytometry at the level of the cell membrane receptor expression.

- Have the authors considered performing cell-cell interaction analysis to predict which ligand-receptor pairs might be driving the interaction between neutrophils and other effector populations in the spleen? One obvious choice would be neutrophil-B cells to predict which interactions might be driving the depletions observed in the spleen and previously reported in this infection system.

As indicated above, this has been addressed in the new Figure 6 (and Suppl. Fig. s6), as well as the new text paragraphs in the result and discussion sections.

- Although I appreciate the manuscript focusses on neutrophils, and this is well justified, I feel like a deeper characterisation of the other splenocyte populations is required as this would be an incredible resource to the community. Some more detailed analysis such of the various innate and adaptive cell types captured in this experiment would be required.

We hope that the full cell annotations in Fig. 1, and the added data in Suppl. Fig. s2, as well as the new Figures 6/s6 will satisfy the curiosity of the reviewer. At the same time however, we fully concur with the idea that the effect of *T. brucei* on every single immune cell population merits an in-dept study on its own, accompanied by cell-immunology validation, as we have done here for the neutrophil-B cell interactions. This would however go beyond the target (and word-count) of this particular publication.

- Line 106: The authors state that infection does not impact the levels of neutrophils in blood and BM, but according to figure 1b there is a significant difference between groups, however small. Can the authors comment?

We agree with the reviewer, and have rephrased the statement (lines 103-105), pointing to the fact that the increase in spleen (SP) neutrophils is still nearly 10 higher than the increase observed in bone marrow (BM) and blood. In Fig. 2 (and Suppl. Fig. s4), we have now also added a newly performed BM scRNA-seq analysis that confirmed the limited increase in BM cell numbers during infections. Indeed, the new analysis shows that the increase in BM neutrophils is minor, that is occurs to roughly the same extent in all BM neutrophil sub-populations, and that no clear infection-associated alterations of gene expression patterns are observed. Despite the editorial comment that no additional scRNA-seq data would be required, we felt that making this data available to the community might help future endeavors to continue the work on trying to better understand the full pathology profile associated to trypanosomiasis in general.

- Regarding figure 1C, the resolution used for dimensionality reduction and UMAP projection does not seem to be included in the methods. Also, it is unclear how the various basic scRNAseq parameters compare before and after filtering. This information should be made available as a supplementary file prior to acceptance of this manuscript.

The information that was already available in the original Materials & Methods section, has now been completed with Supplementary Table s3, covering all QC and other bioinformatics procedures (see also line 634).

- Regarding figure 1C, it would be more informative to relabel the different clusters as the proposed cell IDs (for instance, based on the SingleR analysis) to increase the clarity of what the various cell clusters are. This is provided in figure S5a, but it needs to be made clearer in 1C, as it looks as though all these clusters as neutrophil-related.

We fully agree with this comment and have addressed this in the modified Fig. 1 (and Suppl. Fig. s2) and detailed description in the figure legends.

- Figure 2A, the word "UMAP_" is missing a number.

We apologize for the text-box formatting issue (that also occurred in Fig. 2d). The textbox with has been enlarged, now showing the '1', previously hidden behind Fig. 2b.

- In figure 2, can the authors include the maker genes for each of these populations (or at least the ones referenced in the text – i.e., *Cebpe*, *Ccl6*, *Il1b*) as feature or violin plots? It would help with clarity.

The request has been addressed by (i) marking all key genes that are mentioned in the text with a figure marker (*), color-coded when useful, and (ii) adding supplementary violin plot figure (Suppl. Fig. s3) for the key DEG markers allowing neutrophil population sub-clustering, illustrating 2 key makers per cluster, (8 in total) including *Cebpe*, *Ccl6* and *Il1b* (see also line 112 in the new text).

- In figure 2C, it is not clear whether the N1 population is not present in the naïve samples, or whether it was excluded? Can the authors comment this on the text? Is it possible that N1 appears as a consequence of emergency granulopoiesis?

Indeed, N1 is virtually absent (not excluded) in naïve mice, as shown in Figure 2d (pie chart). The emergence of proliferation competent preNeu precursor N1 subpopulation during the trypanosomiasis-induced inflammation is indeed most likely the result of an emergency granulopoiesis. We have now indicated 'N1' nomenclature more clearly in the discussion (line 349-351).

- Line 135 - 148: Can the authors reference the relevant figures showing the genes listed in the text? It is not immediately obvious.

To satisfy this question, we have paid attempted to include the figure number to the text, every time a specific gene is mentioned. In addition, we have highlighted genes with a marker (*) in the figures, color-coded when useful, starting with Fig. 2f.

- In figure 3a (and throughout) the authors could include an asterisk or arrowhead on the genes discussed in the text to highlight them. This is to facilitate the reading and interpretation of the figures.

As indicated above, we have included (*) indications. We hope this has satisfied the reviewer's request.

- Stats in figure 3e are missing.

Throughout the document, all non-significant events were left 'unmarked' in the first submission of the paper. We have now added 'ns' to all other figures, where this was an issue.

- Is there a corresponding quantification of the results presented in figure 3f?

We have replaced the two flow cytometry plots by a bar/line figure showing that as total neutrophil numbers increase at 14 dpi, the proportion of apoptotic cells within the overall population does not increase (as was previously shown by the lack of % increase of positive cells in the flow cytometry plot). The wording describing the figure has been adapted in line 173).

- Line 204: "none of the 7 markers used, shows" needs rewording

This issue has been corrected (lines 183-195).

- Figure 4e – can the authors comment on the specificity of circulating levels of LCN2 and LTF as a proxy for neutrophil function?

In fact, we cannot. Circulation levels of these molecules most likely involve the macrophage compartment of the liver, but this is speculative at this stage as we have not investigated this. Hence, we would prefer to stay clear from formulating a 'hypothesis' that is not based on experimental data.

- Line 259: has this been observed in other conditions? If so, probably worth referencing here.

To comply with the request, two references have been added to the text (45, 46).

- Figure 6b is missing stats on BM results.

As indicated above, the indication 'ns' has been added to all figures that include non-significant differences between data sets.

- Line 283: can the authors include this datum somewhere in the figure? I think it is important.

This question relates to the question of reviewer 1, asking for actual TIMP levels. We hope that including the new Suppl. Fig s7 and the referral in the text (line 301-308) answers this question. We would like to highlight however that in the initial version of the manuscript, as well as in the main figure body of the revised version, we emphasize on the MMP/TIPM ratio, as this is the factor that determines the biological potential of the protease activity. In reply to a question from 'Reviewer 1' we have included the full data on both TIMP-1 and TIMP-2.

Discussion

- As mentioned above in the results sections, I wonder whether the N1 splenic population can be the result of emergency granulopoiesis? I appreciate this would happen in the BM, but it might be a signature derived from this process. Can the authors comment about this possibility in the discussion?

Besides commenting (line 348-351 and 386-395), we have now added a new sub-figure in Figure 2 (as well as new data in the Suppl Fig s4), with newly obtained data on the BM 14 dpi situation, showing that actually there is no major modulation of neutrophil subpopulations in the BM. We hope that by doing so, we have further increased the value of the manuscript.

- How can the authors reconcile the changed transcriptional landscape of mature neutrophils, as stated in line 383, with their so far known lack of transcriptional activity?

This is a very interesting remark, as one might argue that our new finding questions one of the dogma's put forward about mature (N4) neutrophils. However, it's important to acknowledge that the expanding N4 population we have identified here, is an infection-induced inflammatory population, not the relatively inert N4 population found in naïve non-infected mice. Two other recent transcriptomic studies that also pioneered the identification of neutrophil subpopulation clustering, using COVID-19 and sepsis models. Hence, in the end we hope that our result will spark a continued interest in infection-induced neutrophils, and will lead to the broader realization that mature neutrophils are not *de facto* transcriptionally inactive. They can be inactive, under homeostatic conditions, but as with other immune cells, they can respond to particular 'challenge' conditions, in this case confrontation with extracellular eukaryotic parasites. Our results show that the mammalian host immune system has an amazing capacity to be flexible, and respond to pathogens in ways that have not been fully described yet in the past. Our results also show that the combination of transcriptomic research, in combination with *in vivo* validation, provides a powerful new tool to unravel immune mechanisms that have remained undisclosed so far.

We once again want to thank this reviewer for the very thorough reading and the change we have been given to further improve the value of our manuscript by tackling the questions and comments outlined above.

Reviewer 3:

In this elegant study, authors have used the single cell RNA-seq in conjunction with the current knowledge to show the impact of neutrophils expansion in context of *T. brucei* infection and spleen pathology. The story of the manuscript is tightly and elegantly presented.

We thank the reviewers for expressing his appreciation about the quality of the paper and its presentation.

My main concern is the number of mice used and if data are reproducible considering few mice are used. In majority of figures authors have added a statement that presented data is from 3 mice/experiment with 1-2 experiments. Three mice are not sufficient for power or significance analysis and draw conclusions.

We would like to argue that the power of the significance of all findings combined, lays first of foremost in the fact that all ideas put forward based on scRNA-seq analysis, are independently verified and validated with cellular and/or protein measurement assays. In the original submission, this was done with at least two independent validation experiments for every scRNA-seq based biological primary finding. For the resubmitted version, key validation experiments have now been confirmed with a 3rd repeat. Parts of the legend in Fig. 1 and Fig. 3 have been adapted accordingly. Based on the outcome of these repeats (none of them gave a result that deviated from the results present in the first submission), we are confident in the conclusions that have been drawn.

As additional remark, we can flag that in the meantime we have also independently repeated the spleen scRNA-seq experiment, with 3 mice for each individual group, confirming all the observations that were represented in the initial version of our manuscript.

Considering the pathology of *T. b. infection* is in brain, it would be nice to see the effects of splenic depletion of neutrophils on brain pathology. Does it change?

We agree with the reviewer that the effect of systemic neutrophil depletion (not just the spleen) on brain pathology would be an interesting avenue to pursue, but it goes beyond the goal of this study. A recent study by Quintana et al. (Nat. Commun. doi:10.1038/s41467-022-33542-z, 2022) showed a detailed transcriptomic brain analysis for *T. brucei* infection, but no neutrophil signature was apparent. In addition, a second paper by De Vlaminck et al. (Immunity 8, 2085-2102.e9, 2022) also did not show neutrophil-related data. However, taken the observation that (systemic) MMP-9 is a signature marker for severe HAT pathology, we hope that the work presented in this paper can inspire others, working on trypanosomiasis-associated brain pathology, to pay detailed attention to possible neutrophil mediated effects in the future.

REVIEWERS' COMMENTS

Reviewer #1 (Remarks to the Author):

The authors have addressed all the concerns.

Reviewer #2 (Remarks to the Author):

I have now gone over the revisions for this manuscript and I am pleased to see that my suggestions were thoroughly considered and incorporated. It is indeed a pleasure to read the revised version and will certainly encourage us to look deeper into the role of innate immune cells, in particular neutrophils, in the context of tissue pathology, in particular the brain.

I commend the authors for their work and hence endorse this manuscript for publication.

Reviewer #3 (Remarks to the Author):

Authors have significantly improved the revised manuscript. I have no further comments